# Double-Edged Role of Resource Competition in Gene Expression Noise and Control

*Hanah Goetz, Austin Stone, Rong Zhang, Ying-Cheng Lai, and Xiao-Jun Tian\**

Despite extensive investigation demonstrating that resource competition can significantly alter the deterministic behaviors of synthetic gene circuits, it remains unclear how resource competition contributes to the gene expression noise and how this noise can be controlled. Utilizing a two-gene circuit as a prototypical system, a surprising double-edged role of resource competition in gene expression noise is uncovered: competition decreases noise through introducing a resource constraint but generates its own type of noise which we name as "resource competitive noise." Utilization of orthogonal resources enables retainment of the noise reduction conferred by resource constraint while removing the added resource competitive noise. The noise reduction effects are studied using three negative feedback types: negatively competitive regulation (NCR), local, and global controllers, each having four placement architectures in the protein biosynthesis pathway (mRNA or protein inhibition on transcription or translation). The results show that both local and NCR controllers with mRNA-mediated inhibition are efficacious at reducing noise, with NCR controllers demonstrating a superior noise-reduction capability. Combining feedback controllers with orthogonal resources can improve the local controllers. This work provides deep insights into the origin of stochasticity in gene circuits with resource competition and guidance for developing effective noise control strategies.

H. Goetz
School for Engineering of Matter, Transport and Energy
Arizona State University
Tempe, AZ 85287, USA

A. Stone, R. Zhang, X.-J. Tian
School of Biological and Health Systems Engineering
Arizona State University
Tempe, AZ 85287, USA
E-mail: xiaojun.tian@asu.edu

Y.-C. Lai
School of Electrical, Computer and Energy Engineering
Arizona State University
Tempe, AZ 85287, USA

Y.-C. Lai
Department of Physics
Arizona State University
Tempe, AZ 85287, USA

## 1. Introduction

The last three decades have witnessed increasing exploitation of synthetic gene circuits in clinical applications for diagnostics and therapeutics. In a gene circuit, the competition over transcriptional and translational resources between multiple modules is a universal phenomenon and has been demonstrated to play a significant role in regulating the deterministic behaviors of the synthetic circuits.[1–3] For example, mechanisms through which resource competition can alter the means of circuit species and even completely change the steady states of the dynamical systems has been studied.[4–6] While gene circuits can exhibit certain deterministic behaviors to some extent, they are intrinsically stochastic, and this can significantly affect the circuit function. In fact, noise is one of the fundamental factors that limit the performance of synthetic gene circuits. Such impairment of function has been noted as far back as the repressilator circuit,[7] where fewer than half of the cells showed highly variable oscillations with large variability in period and amplitude. Gene expression noise is one of the sources of uncertainty that can lead to circuit failure.[8] That is, noise, as one of the notorious issues, reduces the forward engineerability of synthetic gene circuits and impairs circuit performance. In view of the importance of resource competition in circuit dynamics, it is of fundamental interest to investigate how the competition affects or contributes to the stochastic nature of the circuits. Understanding the origin of stochasticity in gene circuits is important not only for better understanding intracellular dynamics but also for advancing gene circuit engineering.

Given that noise in gene circuits can have deleterious effects on their predictability and forward-engineerability, methods of controlling and mitigating circuit noise are of paramount interest. In general, the effects of resource competition can be attenuated in orthogonal resource systems and via incorporation of negative feedback control. For example, orthogonal ribosomes and RNA polymerases (RNAPs) have been engineered for creating separate resource pools for genes,[9–13] and various circuit controller topologies have been exploited to mitigate the burden that the components have on each other without affecting the overall function of the circuit.[14–20] The existing controllers utilize some sort of negative feedback topology to

repress circuit outputs when the synthetic circuit begins taking up more than its fair share of the host's resources. Three categories of such controllers have been studied: global,[21] local,[17] and negatively competitive regulatory (NCR) controllers.[20] While these control strategies have been demonstrated to reduce resource competitive effects, it is unclear whether they can reduce gene expression noise and, if so, which architecture represents the optimal design for noise control.

Given the universal existence of resource competition in gene circuits and considering that traditional protein expression models do not take into consideration resource constraints, it is of fundamental importance to uncover and understand the competition-induced noise and characterize the stochastic behavior for synthetic biological constructs. As feedback control serves to reduce noise, it is also imperative to evaluate the efficacy of different control strategies to identify the optimal one. This paper addresses these issues. First, we analytically compared the noise in the idealized scenario in which resources are unlimited with the realistic case subject to the constraint of resource competition. This analysis leads to a new type of noise derived from resource competition and a reduction in noise caused by the resource limitation constraints, revealing a striking double-edged effect of resource competition on noise behavior. We then analyzed how the addition of orthogonal resources can remove the competition-induced noise. Finally, we compared the noise reduction performance of three general types of negative feedback controllers: NCR, local, and global, as well as each of their four placement subtypes. We found that global controllers are not effective at reducing noise and often can even increase noise, but combining negative feedback controllers with orthogonal resources can improve the local controllers and make the transcriptional inhibition strategies more effective. Our results provide unprecedented insights into the origin of stochasticity in synthetic gene circuits as well as into developing effective noise control strategies for resource competitive systems.

## 2. Results

### 2.1. Double-Edged Effects of Resource Competition on Gene Expression Noise

We considered a circuit with two identical yet independently regulated genes in the same cell (represented by GFP and RFP), as shown in Figure 1a, which is a prototypical circuit for modeling and characterizing gene expression noise[22] and resource competition.[4,19] While the competition is not significant with only one copy of the two genes,[22] it can become significant for systems utilizing high-copy plasmids which is sometimes required for proper circuit function. As in previous models for gene expression noise,[23–30] we considered transcription for the production of gfp and rfp mRNAs, translation for the production of proteins GFP and RFP, and the degradation of both mRNAs and proteins. A feature of the existing models is that the transcription rate per gene and the translation rate per mRNA are constants, which are idealized. Going beyond the existing studies, we incorporated the competition for shared transcriptional and translational resources between the two genes into our model, which makes the transcription and translation fluctuate dynam-

ically as in real gene circuits. Resource competition between multiple genetic nodes in the same cell can have significant effects on the deterministic behaviors of the circuit. For example, the sharing of resources can create indirect inhibition links between the genes (indicated by the dashed lines in Figure 1b) because, when one genetic module pulls from the resource pool, there are fewer resources available to the other module.

We began by constructing two mathematical models for the two-gene circuit, one with unlimited resources (without competition) and another with competition, with details given in Section A, Supporting Information. Here, for simplicity, we considered the identical genes with same transcription/translation rates and mRNA/protein degradation rates, and did not consider the folding/maturation difference between the proteins and simply used them for the communication of the work. In the model with unlimited resources (UR model), the translation rate in each module depends linearly on the concentration of its own mRNA, as in previous models.[23,25,26,28–30] In the model with resource competition (RC model), the transcription and translation rates depend on the concentrations of all gene copies and mRNAs in the system, respectively.[6,20] For simplicity, we did not include extrinsic noise from fluctuations in other cellular components of the system. That is, we did not consider the fluctuation of the copy numbers of the transcriptional resource (RNAPs) and translation resource (ribosomes). To fairly compare the stochastic behaviors of the RC and UR models, we rescaled the transcription and translation rate constants in both models to ensure that they have the same mean values of the mRNAs and proteins. We used the standard Gillespie method[31] (Section B, Supporting Information) to generate stochastic trajectories, as shown in Figure 1c, where the peaks in the GFP level more often correspond with valleys in the RFP level in the RC case. Likewise, peaks in RFP correspond with valleys in GFP. That is, the two proteins fluctuate in an anticorrelated fashion. This anticorrelation stems from the coupling of resource competitive and stochastic dynamics. As one protein by chance experiences a stochastic increase in expression levels, fewer resources are momentarily left for the opposing gene to self-express. This phenomenon is confirmed with the 2D GFP/RFP probability distribution obtained as a solution to the system's master equation (Section B, Supporting Information), as shown in Figure 1d, where the deep blue color represents regions of low probability, while yellow represents the regions of the highest probability. It can be seen that a high GFP level mostly corresponds to low RFP levels and vice versa. This is consistent with recent findings of the anticorrelation between two independently regulated gene/modules.[4,6,14,19]

In the UR model, stochastic trajectories and their probability distribution indicate that expressed GFP is not related to the expression of RFP, as shown in Figure S1a,b, Supporting Information. That is, the two genes remain completely unconnected given that the inhibitions from resource competition are not present. We compared the expression distribution of GFP under conditions with unlimited and limited resources and found that the resource competition narrows the GFP distribution, as shown in Figure S1c, Supporting Information. This result implies that resource limitation provides the benefit of noise reduction despite the fact that it also leads to anticorrelation between the expression of the two genes when compared to the unlimited resources case.

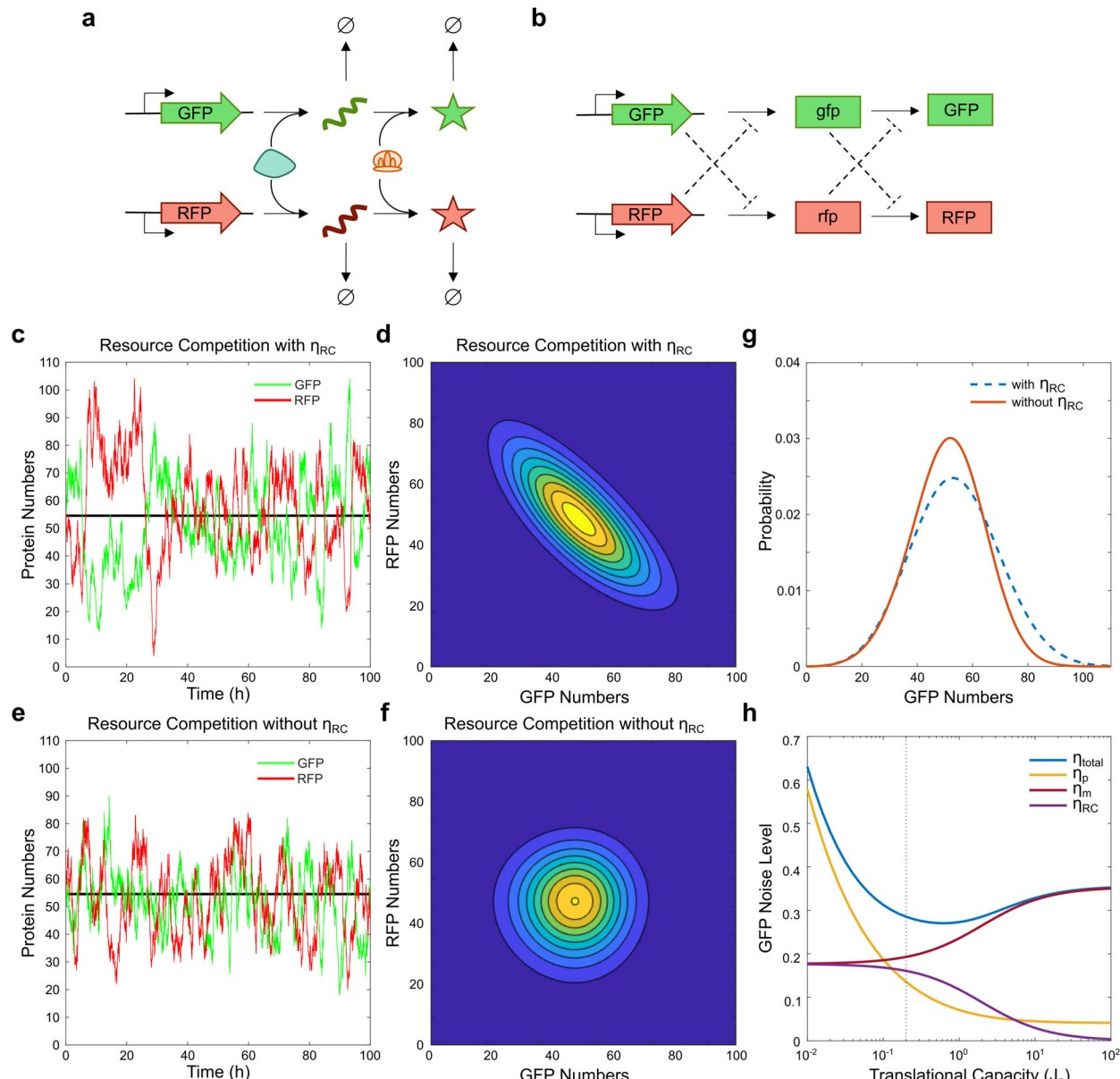

**Figure 1.** Double-edged effects of resource competition on gene expression noise. a) Schematic diagram of GFP and RFP expression through shared transcriptional resources (RNAPs) to create mRNAs and translational resources (ribosomes) to create proteins. b) Diagram illustrating how resource competition creates inhibition between the circuit modules. c) Gillespie stochastic trajectories of GFP (green trace) and RFP (red trace) expressions (correlation coefficient = −0.79). d) Distribution of GFP and RFP expression levels obtained by solving the master equation in the two-gene circuit coupled by resource competition. e,f) Gillespie stochastic trajectories of GFP (green) and RFP (red) expression (correlation coefficient = −0.01) and distribution of GFP and RFP expression levels in the two-gene circuit without RC noise ($\eta_{RC}$, defined as the noise from the fluctuation of the other mRNA due to resource competition). The horizontal black lines in (c) and (e) indicate the protein mean. g) Distribution of GFP expression levels in the case with $\eta_{RC}$ (blue dashed curve) and without $\eta_{RC}$ (solid red curve). h) FDT analytical solutions of the dependence of the total protein noise ($\eta_{total}$, blue curve), noise from the stochastic birth/death of protein ($\eta_p$, yellow curve), noise from the fluctuation of its own mRNA ($\eta_m$, maroon curve), and noise from the other mRNA ($\eta_{RC}$, purple curve) on the translational capacity $J_p$ of limited resources in the host cell for the synthetic gene circuit. Vertical line represents GFP noise levels for the $J_p$ value used in previous panels.

From simulations, we noticed that, due to resource competition, the fluctuations of the mRNA of one protein can contribute to the noise of the other protein. We defined the noise from the fluctuation of the opposing mRNA due to resource competition as resource competitive noise (RC noise), denoted as $\eta_{RC}$. This additional noise can be eliminated by setting the concentration of one mRNA to a constant (e.g., its mean) in the production rate of the other protein, thereby keeping the fluctuations of one gene's expression from being a factor in the opposing gene's expression. By so doing, we were able to determine the contribution of the resource competition to the total noise. Once eliminated, the anticorrelation disappears. That is, protein expression peaks

in the stochastic trajectory can align with either the opposing protein's peaks or the valleys, as shown in Figure 1e. The 2D GFP/RFP probability distribution now becomes circular, as shown in Figure 1f, in contrast to the ovular distribution in Figure 1d. It can be seen that high expression areas are now confined to a smaller region of the phase space than in the original RC case. This difference can also be seen in the 1D GFP probability distribution, as shown in Figure 1g, where the distribution in the RC case (dashed blue curve) is lower and wider than that in the case where the fluctuations of one gene's expression to the other have been eliminated (solid red curve). The results in Figure 1c–g thus indicate that the extra noise included in the RC case can be attributed entirely to resource competition.

To better see the double-edged effects of resource limitation on the noise levels, we derived the analytical expressions of the GFP noise using the fluctuation-dissipation theorem (FDT) for the two models, which yielded the decomposition of the total GFP noise (Section D, Supporting Information). For the RC system, the square of the GFP total noise is given by

$$
\eta^2_{\text{GFP, total}} = \overbrace{\frac{\sigma^2_{p_1}}{\langle P_1 \rangle^2}}^{\text{(total GFP noise)}^2}
$$

$$
= \underbrace{\overbrace{\frac{1}{\langle P_1 \rangle}}^{\text{(GFP birth/death noise)}^2\ (\eta^2_p)}}_{\text{low copy fluctuation}}
$$

$$
+ \overbrace{\underbrace{\frac{\sigma^2_{m_1}}{\langle M_1 \rangle^2}}_{\text{gfp mRNA noise}} \times \underbrace{H^2_{21}}_{\text{static-susceptibility}} \times \underbrace{\frac{1/\tau_2}{(1/\tau_1 + 1/\tau_2)}}_{\text{time-averaging}}}^{\text{(propagated noise from gfp mRNA)}^2\ (\eta^2_m)}
$$

$$
+ \overbrace{\underbrace{\frac{\sigma^2_{m_2}}{\langle M_2 \rangle^2}}_{\text{rfp mRNA noise}} \times \underbrace{H^2_{23}}_{\text{static-susceptibility}} \times \underbrace{\frac{1/\tau_2}{(1/\tau_3 + 1/\tau_2)}}_{\text{time-averaging}}}^{\text{(propagated noise from rfp mRNA)}^2\ (\eta^2_{RC})} \quad (1)
$$

where the first two noise terms are the same as the ones in the UR case (Equation (41) in Section D, Supporting Information). In particular, the first noise term represents the stochasticity from the random birth/death of protein ($\eta_p$), which depends on the average number of GFP proteins. The second term is due to the fluctuations of the gene's own mRNA ($\eta_m$), which depends on the gfp mRNA noise and the contribution of gfp mRNA to its translation quantified by the susceptibility factor $H_{21}$. The noise decomposition for the UR system with two noise terms is consistent with that revealed by previous studies.[23,32,33] The last term in Equation (1) is specific to the RC system, which signifies the stochasticity from the other mRNA and depends on

the rfp mRNA noise and the relative inhibition strength of gfp translation by rfp mRNA as characterized by the susceptibility factor $H_{23}$.

Using this analytical solution, we studied how the GFP noise depends on the parameter $J_p$, which represents the translational capacity of limited resources in the host cell for the synthetic gene circuit. We found that the total GFP noise in the RC system first decreases then increases slightly with $J_p$, as shown in Figure 1h. This is due to the double-edged effects of resource competition on the noise: the noise reduction effect due to the resource limitation and generation of RC noise.

It is worthy to note that RC noise fraction actually depends on the translational rate $k_p$ and the resource capacity $J_p$. As shown in Figure S2a, Supporting Information, the RC noise fraction increases with the translational rate up to more than 30%. Interestingly, it shows a nonmonotonic dependence on $J_p$ with a maximum at the middle, consistent with the dependence of the full width at half maximum of the GFP distribution on $J_p$ (Figure S2b, Supporting Information). The underlying reason for the low RC noise fraction at high or low $J_p$ is that resource competition is not significant with a large value of $J_p$ (high resource level), while the noise from the birth/death of protein (first noise term) is more dominant due to the low copy number of proteins with a small value of $J_p$ (low resource level).

To confirm that resource competition can lead to noise reduction, we compared the dependencies of the GFP noise on $J_p$ in RC and UR systems, as shown by the solid light purple and blue dashed traces in Figure S1d, Supporting Information. It can be seen that GFP noise is always smaller in the RC case, which is also proved mathematically (Section E, Supporting Information). As $J_p$ increases its value, GFP noise in the RC case approaches the noise level in the UR case. This is reasonable because, as $J_p$ approaches infinity, the terms in the RC model are reduced to those in the UR model, and the limitation on resources gradually disappears.

Figure 1h and Figure S1d, Supporting Information also demonstrate how the noise composition changes with the resource availability. Specifically, the noise from the protein birth/death ($\eta_p$) decreases with the resource availability (the yellow curve in Figure 1h and Figure S1d, Supporting Information), due to the increased protein mean (the red curve in Figure S2c, Supporting Information). It is also noted that $\eta_{RC}$ and $\eta_m$ depend on $J_p$ through $H_{23}$ and $H_{21}$, which decrease and increase with $J_p$, respectively (Equations (35) and (36)), as shown in Figure 1h. That is, the noise due to the stochasticity from its own mRNA increases with the resource availability in the RC system (the maroon curve in Figure 1h) due to the continuous relaxation of the resource limitations on translation, and approaches the level in the UR case, which does not change with translational capacity (Equation (1) with $H_{23} = 0$ and $H_{21} = -1$ as detailed in Section D, Supporting Information and displayed by the maroon curve in Figure S1d, Supporting Information). The RC noise decreases with the resource availability (the purple curve in Figure 1h), as the inhibition of one mRNA on the translation of the other mRNA decreases with increasing translational capacity. Taken together, our finding shows that resource limitation can decrease gene expression noise but in turn creates a new type of noise, resulting in a remarkable double-edged effect on gene expression noise.

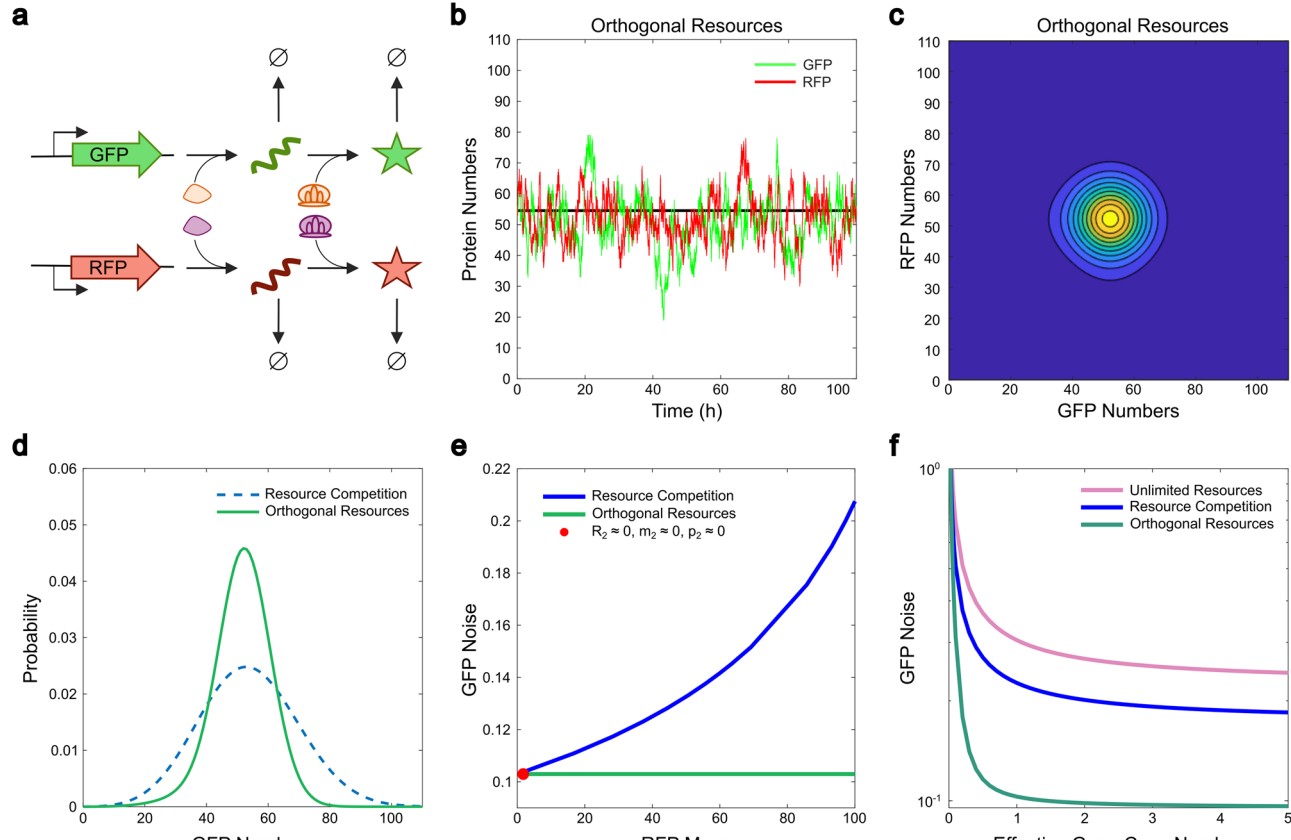

**Figure 2.** Elimination of resource competition noise through orthogonal resources. a) Diagram illustrating how the use of orthogonal resources eliminates the competition for transcriptional and translational resources. b) Gillespie stochastic trajectory of GFP (green curve) and RFP (red curve) expression in a two-gene circuit with orthogonal resources (correlation coefficient = 0.02). The horizontal black line indicates the protein mean. c) Distribution of GFP and RFP expression levels in a two-gene circuit with orthogonal resources obtained from the solutions of the master equation. d) Distribution of GFP expression levels in a resource competition system (blue dashed curve) and an orthogonal resources system (green curve). e) FDT analysis revealing the dependence of the GFP total noise levels on RFP mean in a system with resource competition (blue curve) and orthogonal resources (green curve) using the latter as the base. f) FDT analysis giving the dependence of the protein total noise levels on the effective gene copy number in a system with unlimited resources (pink curve), resource competition (blue curve), and orthogonal resources (green curve).

## 2.2. Elimination of Resource Competition Noise through Orthogonal Resources

Orthogonal resources such as orthogonal RNAP and ribosomes have been developed to reduce unwanted couplings in gene circuits due to the competition for the host transcriptional and translational component.[9–13] We set out to study how the introduction of orthogonal resources affects noise levels. With their addition into the two-reporter system, gfp and rfp now pull from two separate pools of RNAP and ribosomes, as shown in **Figure** 2a. Here, we assume that the orthogonal system uses much less resources than the synthetic gene circuit and thus did not consider the potential resource competition by the orthogonal system. As a result, the inhibition links between the two genes caused by resource competition are removed while the resource constraint remains. We hypothesized that this would retain the noise decreasing effects of resource competition while nullifying the resource competitive noise.

We constructed a model for the orthogonal resource (OR) system (Equations (11) and (12) in Section A, Supporting Informa-

tion) and compared its noise behavior with the RC system by rescaling the transcription and translation rate constants in the OR model to keep the same means of the mRNAs and proteins in the two models. The stochastic trajectories reveal that the expression levels of both proteins are no longer anticorrelated and fluctuate closer to the mean, as shown in Figure 2b. Accordingly, the 2D GFP/RFP probability distribution is more compact near the mean, as shown in Figure 2c. The probability distribution for the OR case (the solid green curve in Figure 2d) is narrower than the RC case (the dashed blue curve in Figure 2d), suggesting that utilization of orthogonal resources can reduce the protein noise levels. An analytical solution of the protein noise shows that the resource competitive noise has been removed from the system (Section D, Supporting Information).

We then studied how the GFP noise levels in the RC and OR systems change with the RFP mean by increasing the effective RFP gene copy number. By rescaling the parameters in the RC system with the OR system as the base, we make the means of the mRNAs and proteins in the RC system the same as the OR system. We found that, when there is no RFP, the noise levels in

the two systems are equal, as shown by the red dot in Figure 2e. As the RFP mean increases, the GFP noise level in the OR system is constant given that its two components do not change with RFP mean (Figure S3a, Supporting Information), but in RC system the total noise increases because both the noises from the fluctuation of two mRNAs ($\eta_m$ and $\eta_{RC}$) increase although the noise from the birth/death of GFP ($\eta_p$) is a constant (Figure S3b, Supporting Information), thus making the difference in noise level between two systems increase with RFP mean (Figure 2e). Note that the maximum RFP mean reached by increasing the copy number is due to the saturation mediated by resource limitation, as shown in Figure S3c, Supporting Information. This conclusion holds true if the plasmid copy numbers increase for both genes. The total noise decreases and reaches a saturation floor, as shown in Figure 2f, which is consistent with the previous findings.[34–37] Nonetheless, the noise levels in the UR system are always the largest, suggesting that utilizing orthogonal resources is a good strategy to eliminate the contribution of resource competition to gene expression noise.

### 2.3. Control of Gene Expression Noise by Negatively Competitive Regulatory (NCR) Controllers

Negative feedback has been utilized extensively to mitigate the effects of resource competition[14–20] and reduce gene expression noise.[38–47] Some theoretical analyses have provided us a fundamental limitation to the noise suppression through negative feedback loops.[48,49] While cellular systems in nature may have evolved complicated regulatory networks to operate close to this fundamental limit, it is still unclear whether different types of negative feedback controllers perform similarly for noise reduction of the synthetic gene circuits, especially under the context of resource competition. We systematically studied the noise attenuation ability of these negative feedback controllers in the context of resource competition. Previously, we proposed a controller topology for combating resource competition effects—negatively competitive regulatory (NCR) controller,[20] as schematically illustrated in **Figure 3**a. Briefly, in addition to their output proteins, each competing module also creates an sgRNA which is inhibitory towards the module that produced it. However, these sgRNAs cannot initiate inhibition until they complex with an inhibitory CRISPR moiety (e.g., dCas9), which are drawn from a fixed pool. The resulting inhibitory complexes can then initiate inhibition of their respective modules. This is an example of mRNA-mediated inhibition of transcription. Such a controller topology can be generalized to be placed at any one of four places in the protein production pathway, defined by whether the inhibition is mediated via mRNA or protein and whether the controller targets transcription or translation for inhibition. We henceforth defined four controller subtypes, as shown in Figure 3b: mRNA inhibits transcription (MIX), protein inhibits transcription (PIX), mRNA inhibits translation (MIL), and protein inhibits translation (PIL).

To assess the noise reduction ability of these NCR controllers, we developed a generalized model for these controllers (Supporting Information Section F) and carried out FDT-based numerical analysis (Supporting Information Section G) for all the four subtypes with increasing RFP means for a fixed controller strength.

Here we did not consider the resource competition by expressing dCas9 as we need the dCas9 to be limited so that the two modules could compete over to achieve the negative competition as designed in the NCR controller. For future experimental design, a tunable dCas9 system will be integrated into the genome instead of the synthetic gene circuit plasmid. To compare these controllers fairly, both $S_c$ and RFP start at 0 to represent no controller or no competing module. The maximum of $S_c$ was set to the value when the noise reduction reaches saturation, and the maximum value of $R_2$ was set to the value where the RFP mean reaches its saturation. Figure 3c shows the GFP noise normalized by the base case without any controller. For comparison, Figure S4a, Supporting Information shows the case of non-normalized GFP noise. We found that translation-inhibiting subtypes (MIL and PIL) have the largest noise reduction effects as RFP mean increases but they perform poorly at low RFP means where GFP noise even increases. Transcription-inhibiting subtypes (MIX and PIX) do not reduce noise significantly at high RFP means but are able to decrease noise consistently over all RFP mean values. Importantly, mRNA-mediated controllers (MIX and MIL) outperform those mediated by protein (PIX and PIL) in both the inhibition of transcription and translation cases. Figure 3d shows that normalized GFP noise decreases as the controller strength increases with a fixed large RFP mean but saturates beyond a certain point. Under a smaller or moderate RFP mean, normalized GFP noise in the PIL case can increase, as shown in Figure S4b, Supporting Information, or first decreases then increases, as shown in Figure S4c, Supporting Information with increasing controller strength. These trends hold with other RFP means and controller strength. Figure 3e–h demonstrates how GFP noise changes in the phase plane of RFP mean and controller strength for all controller subtypes, where deep blue regions represent large noise reduction and dark red regions indicate either little noise reduction or an increase in noise. MIL and PIL subtypes have both regions of deep blue but also dark red, indicating their increasing ability of noise reduction with RFP mean but poor starting performance at low RFP mean intervals. Heatmaps of MIX and PIX have neither much deep blue nor dark red region, indicating their ability to consistently reduce noise over the entire RFP mean interval, albeit at moderate levels. It is worth noting, the noise reduction capabilities smoothen out over the RFP range for all these controller types when the protein noise of the entire system (defined as the Pythagorean sum of both GFP and RFP noise) are taken into account as is detailed in Figure S4d–h, Supporting Information. The large increases in noise generated by the translational-inhibiting controllers, MIL and PIL, at low protein mean are buffered by the large decreases in the noise of the other protein.

### 2.4. Control of Gene Expression Noise Using Local and Global Controllers

We investigated local and global negative feedback controllers that have been used previously to mitigate resource competitive effects.[17,21] Briefly, local controllers incorporate separate negative feedback loops, that is, every module in the genetic system has its own negative feedback loop as shown in **Figure 4**a, whereas global controllers use a single shared negative feedback

**2100050 (6 of 13)**

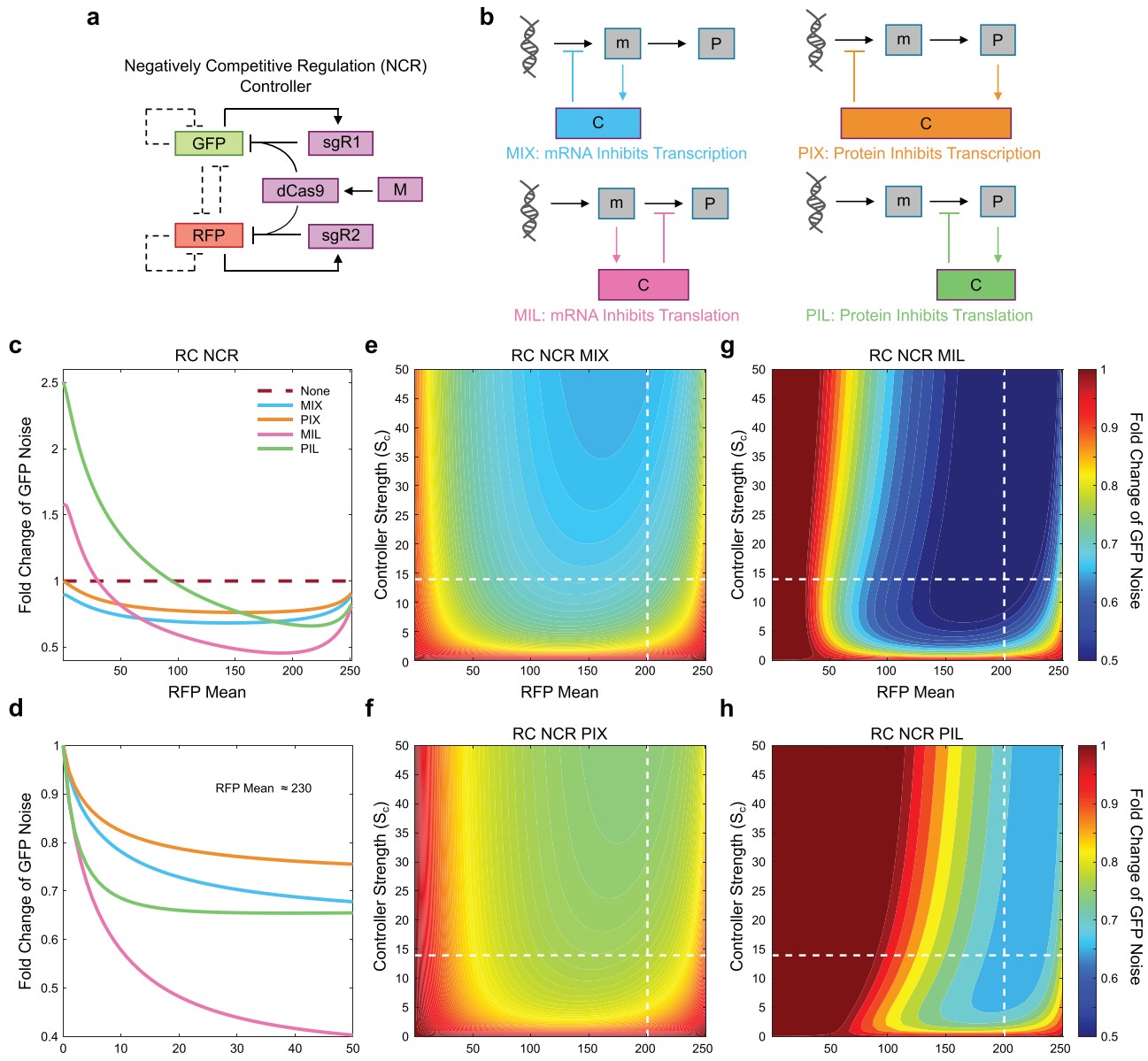

**Figure 3.** Control of gene expression noise by negatively competitive regulatory (NCR) controllers. a) Topology of the NCR controller acting on a two-gene circuit. Each module (GFP, RFP) produces an inhibitory guide RNA (sgR1, sgR2) which is used for self-inhibiting complexes upon binding to a fixed pool of CRISPR moiety (dCas9). The blunt dash arrows show the self-inhibition and mutual inhibition between two modules due to resource competition. b) General controller topologies applied at different positions in the protein production pathway defined by the inhibitory moiety type and the target of inhibition: mRNA inhibits transcription (MIX), protein inhibits transcription (PIX), mRNA inhibits translation (MIL), and protein inhibits translation (PIL). c) FDT analysis demonstrating the normalized GFP total noise level on the RFP mean with NCR controllers applied in a resource competitive system for fixed controller strength $S_c = 14$. The noise levels are normalized to the base case without a controller. d) The dependence of normalized GFP noise on $S_c$ for RFP Mean fixed at 230. e–h) Normalized GFP noise in the phase plane of RFP mean and NCR controller strength for NCR controller subtype e) MIX, f) PIX, g) MIL, and h) PIL. Deep blue color represents a strong decrease in the noise levels with respect to the base case with no controller, and deep red indicates the absence of a significant noise change or even a noise increase over the base case. The horizontal and the vertical white dashed lines represent the controller strength in panel (c) and the RFP mean used in panel (d), respectively.

loop that represses all modules in the circuit, as shown in Figure 4b. To compare the efficacy of these controllers in attenuating noise in the two-gene circuit, we performed FDT analysis for the systems with each controller applied utilizing one of the four placement subtypes. We found that global controllers perform poorly. First, the Global MIX and PIX systems reduce the noise slightly at low RFP means but are barely able to reduce noise at higher RFP means, as shown in Figure 4c and Figure S5a, Supporting Information. Second, the global MIL and PIL systems significantly increase noise for most of the RFP mean intervals, as shown in Figure 4d and Figure S5b, Supporting Information. This finding is consistent with the global controller's inability

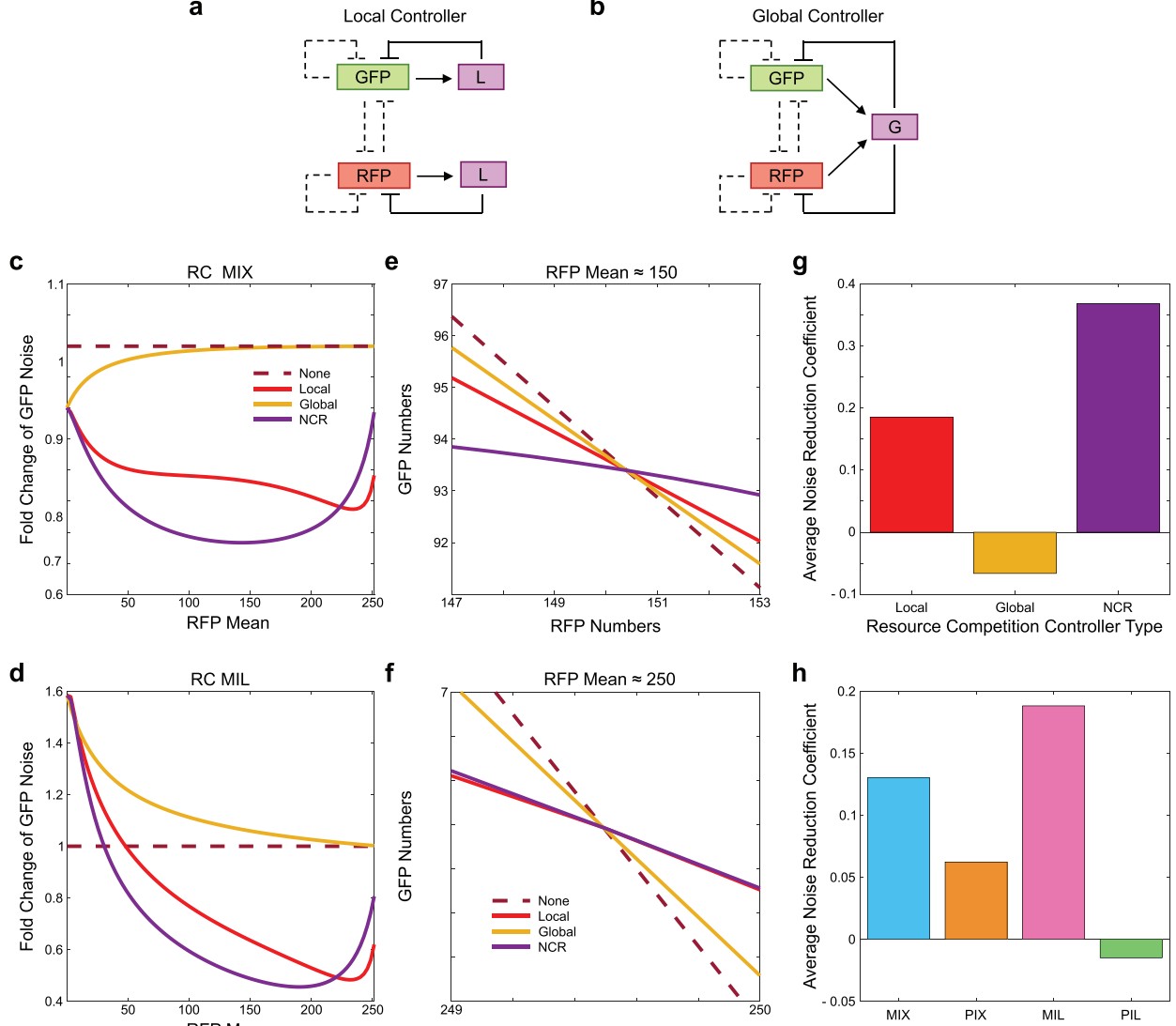

**Figure 4.** Control of gene expression noise using local and global controllers. a,b) Diagrams of the local and global controllers, respectively, acting on the two-gene system. c,d) Dependence of the normalized GFP noise levels with local controllers (red curves), global controllers (yellow curves), and NCR controllers (purple curves), respectively, using MIX and MIL subtypes. The noise levels are normalized to the base case with no controller. e,f) Deterministic dependence of GFP level on RFP level with various MIL controllers applied, where the parameters are fixed to intersect just at one point for RFP mean approximately 150 and 250, respectively. g) Average noise reduction coefficient for local, global, and NCR controller types. h) Average noise reduction coefficient for controller subtypes MIX, PIX, MIL, and PIL.

to attenuate the winner-takes-all resource competitive behavior found previously.[20]

Systems with a local controller perform more poorly than NCR controllers at low/moderate RFP means, as shown in Figure 4c,d and Figure S5a,b, Supporting Information. However, there exists a critical RFP mean value for which Local and NCR cross in efficacy. The value of this cross-point is parameter-dependent, but NCR typically performs better at low/moderate RFP means while a local controller typically performs better at high RFP means. The reason for this cross in controller efficacy can be seen from the GFP versus RFP mean graphs, as shown in Figures 4e,f. As the RC noise in GFP is qualitatively related to the slope at a given point on the GFP vs RFP mean graph, a shallower slope indi-

cates weaker noise. At low/moderate RFP means (Figure 4e), the two-gene system with the NCR controller applied has a rather flat slope, indicating that the NCR controller can nearly decouple the two genes from their resource competition at this point and thus is better at noise reduction. However, at a high RFP mean (Figure 4f), the slopes of the curves with NCR and local controller interchange, with the local controller curve now having the shallowest slope and thus a better noise reduction capability.

To obtain more general and conclusive results regarding which controllers and placement subtypes are optimal for noise reduction, we defined an average noise reduction coefficient as the average fold change in total protein noise (defined as the Pythagorean sum of both GFP and RFP noise) across all protein

means, controller strengths, and/or controller types/subtypes (Section H, Supporting Information). The average noise reduction coefficients are shown in Figure 4g for each controller type and in Figure 4h for all three controller types. These results demonstrate that the global controllers and the PIL controllers on average perform poorly (given their negative noise reduction coefficients) and are generally inappropriate for building an efficacious noise reducing system. Further support for this finding is presented in Figures S5c,j and S6c,j, Supporting Information, where a deep red color emerges for the majority of the phase plane of RFP mean and control strength for the global controller. While Local and NCR systems are both relatively efficacious in attenuating noise, NCR is significantly more consistent in its ability to reduce noise, as shown in Figure 4g. The two best performing placement topologies are the MIX and MIL subtypes, with the MIL placement more consistent in its ability to reduce noise (Figure 4h). We also found that this difference between MIX and MIL increases with increasing $J_p$ values, with MIL outperforming MIX. We found this trend to also be present when orthogonal ribosomes are included (detailed further below). Furthermore, it can be concluded that a system's noise reduction capability in general is determined more by the type of controller than by which controller subtype chosen. This can be seen as the noise reduction coefficients from altercation of controller type (Figure 4g) span approximately 0.36, whereas altercating the controller subtype only results in a smaller span of approximately 0.19 (Figure 4h).

## 2.5. Control of Gene Expression Noise through Combined Negative Feedback Controllers and Orthogonal Resources

Having demonstrated that both orthogonal resources and negative feedback controllers can attenuate noise, we investigated whether their combinations can improve the noise-control capability. We first focused on the combinations of the most effective controllers in the RC system, including local-MIX, local-MIL, NCR-MIX, and NCR-MIL, with orthogonal resources. We then determined whether the application of an OR system can benefit negative feedback controllers for noise reduction. As shown in **Figure 5**a and Figure S7a– c, Supporting Information, the GFP noise normalized to the RC base case without a controller indicates that the use of orthogonal resources consistently benefits local-MIX and NCR-MIX controllers. However, using an OR system is barely beneficial to local-MIL and is deleterious to NCR-MIL, as shown in Figure 5b. The underlying reason is that for the NCR-MIL controller, when the control node from RFP mRNA inhibits its translation, it also promotes the translation of GFP regardless of whether or not GFP level is above or under its average in the OR case, and thus could actually increase GFP noise. In the RC case, on the other hand, it only promotes the translation of GFP when GFP is under its average thus decreasing GFP noise. That is, NCR is designed for combating resource competition, so it may not work well with OR system. The GFP noise normalized to the OR case without a controller reveals that the addition of a local-MIX or NCR-MIX controller is consistently beneficial for the OR system in further attenuating noise, as shown in Figure 5c and Figure S7d– f, Supporting Information. Even though local-MIL and NCR-MIL controllers do not provide any

benefit for the OR system at small RFP mean in reducing noise, the synergy emerges in large RFP mean intervals as shown in Figure 5c. There is thus consistent synergy between the local/NCR controllers and orthogonal resources.

Heatmaps of each of the four controller subtypes over the RFP mean-controller strength plane demonstrate similar behaviors for a range of controller strengths, with the fold changes in GFP noise becoming more pronounced as controller strength increases, as shown in Figure 5d–g. Both combinations of an OR system with Local-MIX and NCR-MIX are the most effective at reducing noise when compared to an RC system without control. However, the OR local-MIL and OR NCR-MIL combinations possess a remarkable ability to reduce noise for higher RFP mean values, but prove disadvantageous at lower RFP mean.

To quantify the general synergy between the negative feedback controllers and orthogonal resources, we calculated the average noise reduction coefficients for all controller types and subtypes in the OR system, as shown in Figures 5h-i. Similar to the system without orthogonal resources, the global controller and the PIL subtype perform poorly, though the global controllers (MIX and PIX subtypes) are a much better contender when added to the OR system (Figure 5h and Figures S5a,b and S7b,e, Supporting Information). Furthermore, the local and NCR controllers perform well, with NCR being on average more consistent in its noise-reduction capability. However, the order of the most efficacious controller subtype is different when applied to an OR system than the RC system, as shown in Figures 5i and 4h, where transcriptional inhibition subtypes (MIX and PIX) demonstrate a better ability to reduce noise than translational inhibition subtypes (MIL and PIL). Figure 5j shows the dependence of the accumulative noise reduction coefficient on the controller strength for the eight best controller and subtype combinations and Figure S8, Supporting Information shows the analysis on the control of total protein noise. The curves in the OR systems have a much higher starting point at small controller strength range due to the strength-independent noise reduction resulting from the orthogonal ribosomal component (Figure 5j). The NCR controllers are always better than local controllers in the RC systems, while in the OR system, the latter are slightly better than the former. This is reasonable as the NCR controller is specifically designed to mitigate the effects of resource competition. The addition of orthogonal resources significantly improves the less efficacious NCR-MIX and local-MIX controllers but does not enhance the maximum noise-reduction capability for the local-MIL and NCR-MIL controllers. This is also validated by the Gillespie simulation (Figure S9, Supporting Information). It is worth noting that, although the noise reduction coefficient of the PIX subtypes is greater than that of MIL subtypes here (Figure 5i), this effect is largely due to the fact that the global-PIX controller is significantly improved (Supplementary Figure S7b). That is the reason we chose to analyze MIL over PIX in the best cases comparison in Figure 5j despite PIX's seemingly higher noise reduction coefficient.

## 3. Discussion

Uncovering and understanding the origin of gene expression noise is a fundamental problem in systems and synthetic biology and has been investigated extensively in the past.[23–30] However,

**2100050 (9 of 13)**

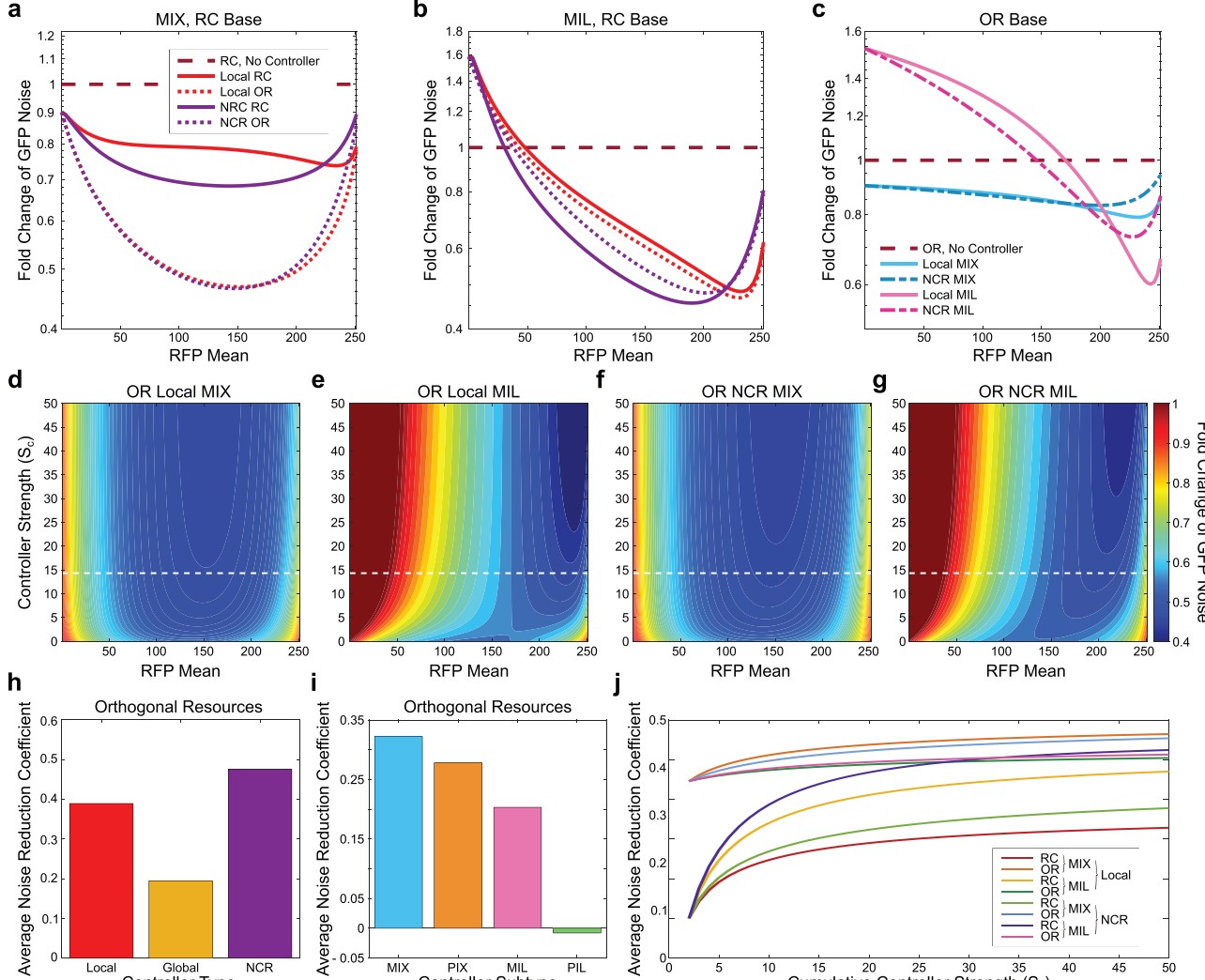

**Figure 5.** Control of gene expression noise through combined negative feedback controllers and orthogonal resources. a,b) Dependence of the normalized GFP noise levels on RFP mean with a local controller (red curves) and NCR controller (purple curves) combined with orthogonal resources (solid curves), respectively, for MIX and MIL controller subtypes. The dashed curves are cases without orthogonal resources. All data are normalized to those of the system without a controller (maroon dashed curve). c) Dependence of the normalized GFP noise levels on RFP mean for the OR system combined with a local controller (solid curves) or an NCR controller (dashed curves) for either the MIX (blue curves) or MIL (pink curves) controller subtypes. All data are normalized to those in the case with orthogonal resources but no controller (maroon dashed curve). d–g) Normalized GFP noise in the phase plane of RFP mean and controller strength, respectively, for local-MIX, local-MIL, NCR-MIX, and NCR-MIL combined with orthogonal resources. Deep blue color represents regions of higher noise reduction. The horizontal white dashed lines represent the controller strength shown in panels (a) and (b). The average noise reduction coefficient for h) different local, global, and NCR controllers with orthogonal resources, and i) different controller subtypes: MIX, PIX, MIL, and PIL with orthogonal resources. j) Dependence of the average noise reduction coefficient on the cumulative controller strength for the cases MIX and MIL of NCR/local controllers applied with or without orthogonal resources.

in all the existing models, unlimited cellular resources are assumed, which is unrealistic for synthetic gene circuits with multiple coactivated modules. Our present work has demonstrated quantitatively that resource competition can significantly affect the noise behavior of synthetic gene circuits. A key finding is that resource competition has a double-edged effect on protein expression noise levels. In particular, resource competition is able to reduce noise by applying resource constraints to the system, in the absence of which the gene expression noise level is maximized. The noise-reduction capability can be attributed to self-inhibition introduced indirectly into the system by resource

competition. However, the competition introduces a new type of noise (resource competitive noise) as it creates anticorrelated links between the gene modules. This double-edged effect makes the dependence of the total noise level on the resource availability strikingly non-monotonic. Incorporation of orthogonal resources can take advantage of this effect to reduce the noise by orthogonalizing the resource constraints on each gene module. This technique keeps the resource constraints but removes the effects of inter-module resource competition, allowing for fewer variables affecting gene expression while still using self-inhibition to keep the expression levels relatively close to the

mean. Development of a completely insulated OR system in the future has the potential to significantly improve the control of resource competition. Current OR systems are not completely orthogonal to the host system and take up additional resources in the host cell, thus it is necessary to also consider other control strategies such as feedback controllers to further improve the control of resource competition.

From a control perspective, negative feedback loops are often used to suppress the noise level in gene expression.[38–47] However, previously none of the methods took into account resource competition. Nonetheless, a number of negative feedback controllers such as local, global, and NCR controllers have been used in the past to negate unwanted effects of resource competition,[14–20] raising the question of whether these controllers can be used to reduce gene expression noise in the circuits with limited resources. Our analysis reveals that global controllers are typically quite ineffective at reducing noise and often can increase it. In a recent work, we demonstrated that global controllers are not efficacious at reducing the effects of winner-takes-all type of resource competition,[20] implying its inability to reduce resource competitive noise. Local and NCR controllers, however, both are efficacious at reducing protein noise, with the latter outperforming the former at low RFP means and the opposite behavior at high RFP means. A useful result is that the controller placement topology within the protein biosynthesis pathway can drastically alter the noise reduction ability of a controller. Particularly, we found that inhibition mediated via mRNA (MIX and MIL subtypes) is more efficacious than inhibition mediated by protein (PIX and PIL subtypes). The reason why MIX and MIL are more efficacious than PIX and PIL is that proteins have three sources of noise while the mRNA noise only depends on its birth/death. In this way, protein has more noise sources than mRNA, which would cause more variance if they were the effectors for the controllers. Combining orthogonal resources systems and negative feedback controllers makes the transcriptional inhibition strategies (MIX and PIX subtypes) more effective than translational inhibition strategies (MIL and PIL subtypes). The conclusions from this work theoretically apply to any organism as long as there exists significant resource competition, which is already observed in both bacteria and mammalian cells.[4,6,14,19]

It is important to note that the large array of negative feedback controller topologies analyzed in this work have biological underpinnings and are not merely theoretical. For example, the MIX, MIL, and PIX subtypes of the local controller have been demonstrated in vivo by various groups,[17,19,50] the local-PIL controller can be constructed via expression of orthogonal, sequence-specific RNA-binding proteins as inhibitory effectors.[50,51] Global controller architectures can be constructed from each of these systems by replacing the orthogonal feedback modules with copies of the same regulator[21] where each module performs the same negative feedback operation but these production pools are shared amongst modules. While the NCR controller type is the newest proposed negative feedback architecture, it has not yet been synthetically constructed in vivo models. In our recent work,[20] we demonstrated how an NCR-MIX controller could be theoretically constructed using inhibitory deactivate dCas systems. It is important that careful consideration should be given to which dCas system is incorporated in such a design as many dCas systems suffer from the problem of strong/irreversible dCas

moiety binding.[52] This NCR-MIX controller assumes a constant production of Cas moiety while individual genetic modules produce sgRNAs that mediate repression. Numerous dCas systems have been reported to demonstrate sequence-specific RNA-binding capabilities, opening up the potential for in vivo MIL constructs for the NCR controller type.[53] Many of these systems also exhibit sequence-specific DNA-binding behavior. However, recently a few systems have been discovered that naturally target RNA such as Csm3, Cmr4, Csm6, and Csx1, the extensive family of Cas13 proteins.[54,55] In addition, complex signal processing functions have been designed utilizing adaptive zinc finger protein complexes (e.g., Bashor et al. demonstrated that a generalized system utilizing sequence-specific zinc-finger proteins (ZFPs) and a clamp composed of PDZ moieties strung together which accepts these ZFPs is capable of performing very complex signal processing[56]). Since complexing between a conserved moiety (PDZ clamp) and sequence-specific moieties (ZFPs) is required to form the inhibitory complexes, it is possible to generate NCR-PIX systems. Furthermore, certain ZFP constructs have been demonstrated to have sequence-specific RNA-binding capabilities: such RNA-binding ZFPs can be potentially paired with PDZ/PDZ-like clamps to open up the possibility of NCR-PIL systems.[57]

In our work, potential sources of extrinsic noise in the system such as the fluctuation of the copy numbers of the transcriptional resource (RNAPs)[37] and translation resource (ribosomes) are not included. It was reported that sharing a common regulator pool could result in indeterminacy of extrinsic noise.[58] It is worth noting that resource competition makes the two-reporter expressions anticorrelated. Significant extrinsic noise that makes two reporters fluctuate in a positively correlated fashion, together with strong resource competition, may lead to a circular 2D probability distribution. Our models do not take into account the bursting of gene expression, which can be simulated with a two-state model[28,59] and can lead to a significant noise at the transcriptional level, especially under the context of a limited level of RNAPs. The contribution of these factors to the noise of synthetic gene circuits needs to be characterized. Furthermore, our noise analysis has been carried out using the most basic resource competitive system: an unregulated, unlinked two-gene system. For future investigation, we intend to look at the noise behavior in more complex and dynamic systems, such as the dual self-activating or cascading circuits that we have analyzed recently.[6,20] The noise from other circuit-host interactions such as growth feedback[60–65] adds another layer of complexity to the stochastic gene expression of synthetic circuits. The stochasticity in cellular growth and its propagation to the synthetic gene circuits is another potential source of fluctuations.[65–68] It may also prove insightful to investigate the noise-reducing effects of other methods for attenuating resource competitive effects, such as incorporating incoherent feedforward loop topologies[14,18,69] or antithetic integral feedback[70,71] into the circuit. It is noted that gene expression noise could be exploited even it is undesirable for the deterministic functions of gene circuits. For example, gene expression noise induces a bimodal response in a positive feedback loop circuit without cooperativity.[72] At the populational level, stochastic phenotype switching due to the gene expression noise gives bacteria an advantage in fluctuating extracellular environments.[73–77]

# ADVANCED SCIENCE NEWS

www.advancedsciencenews.com

# ADVANCED GENETICS

www.advgenet.com

## Supporting Information

Supporting Information is available from the Wiley Online Library or from the author.

## Acknowledgements

This project was supported by NIH grant (R35GM142896 to X.-J.T.). H.G. and A.S. were also supported by the Arizona State University Dean's Fellowship. Y.-C.L. was supported by Office of Naval Research under Grant No. N00014-21-1-2323.

## Conflict of Interest

The authors declare no conflict of interest.

## Author Contributions

H.G. and A.S. contributed equally to this work. X.-J.T. conceived and designed the study. H.G., A.S., and X.-J.T. performed studies. H.G., A.S., R.Z., Y.-C.L., and X.-J.T. analyzed the data. H.G., A.S., and X.-J.T. wrote the manuscript. H.G., A.S., R.Z., Y.-C. L. and X.-J. T. edited the manuscript.

## Data Availability Statement

The data that support the findings of this study are available from the corresponding author upon reasonable request.

## Peer Review

The peer review history for this article is available in the Supporting Information for this article.

## Keywords

noise reduction, global negative feedback control strategy, local negative feedback control strategy, negatively competitive regulation, noise decomposition, resource competition noise, resource competition

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

**2100050 (13 of 13)**
