## [**Supplementary Information**: Record of Transparent Peer Review · Advanced Genetics]

Record of Transparent Peer Review

Double-edged role of resource competition in gene expression noise and control

Hanah Goetz, Austin Stone, Rong Zhang, Ying-Cheng Lai, Xiao-Jun Tian*

*Corresponding

Review timeline:	Date Submitted:	27-Sep-2021
	Editorial Decision:	15-Nov-2021
	Revision Received:	17-Dec-2021
	Editorial Decision:	5-Jan-2022
	Revision Received:	8-Jan-2022
	Accepted:	10-Jan-2022

1 st Peer Review	10/1/2021 to 11/15/2021
-------------------------

Reviewer #1

In this manuscript, Goetz et al. present modeling results focusing on understanding how resource competition (RC) impacts the noise of genetic circuits. They also explore how different noise-reduction controllers perform in the presence of resource competition and when the synthetic proteins are expressed using orthogonal expression machinery. This is an interesting study, and while results haven't been confirmed experimentally, it could help guide the design of future synthetic circuits. The manuscript is generally easy to read and the figures are easy to understand. However, before being ready for publication, the authors should address the concerns listed below.

Major concerns:

1.1 A key point of the paper is that RC both reduces and increases noise and the balance of these two properties results in a non-monotonic dependence of the total noise level on resource availability. Indeed, this point is central to the paper as it defines its title. The authors mention that "We... find that resource competition narrows the GFP distribution, as shown in Fig. 1c". I assume that the authors meant 1g (distribution) and not 1c (stochastic trajectories). If indeed the graph the authors reference is 1g, the GFP distributions are nearly the same between RC and non-RC conditions. The distribution with RC may even be slightly wider, but the widths (full width at half maximum) were not quantified and compared. Regardless of which distribution is wider, the difference is so small that I am not yet convinced of its importance.

1.2 Figure 1f suggests that GFP and RFP expression isn't correlated in the non-RC case. However, Fig. 1e shows clear anti-correlations. The relative shape of GFP and RFP expression is so striking that the anti-correlations are unlikely be a visual artifact. Can the authors explain the discrepancy?

1.3 Where are the values for the parameters used in the model e.g., mRNA degradation rate, transcription and translation rates, etc.? GFP and RFP have different folding/maturation properties; is that in the model? I don't think that the differences in properties between GFP and RFP are important for this story. Given this, why not just model "Protein 1 (P1)" and "Protein 2 (P2)"? If you want to keep using GFP and RFP so simplify the communication of the work, please clarify that they have identical properties in the model.

1.4 There is generally a lack of quantification in the manuscript. For example, on p.8 (first paragraph), the authors say "Accordingly, the 2D GFP/RFP probability distribution is narrower than the RC case", but no quantification (coefficient of variation) is provided. Another example is in the previous sentence, where correlations/anticorrelations are not quantified. Please rectify this for the entire manuscript.

1.5 Fig. 2e. Why does GFP noise increase with RFP expression in the OR case? Shouldn't their expression (and therefore their noise) be orthogonal? Is it an artifact of the scaling method used? If so, I'm not convinced that these results are useful.

1.6 It is not clear whether the differences between controllers (MIL, PIL, MIX, and PIX) are due to differences in their design or the value of specific parameters. a. How robust are the results to their parameters? b. Average plots like Fig. 4g/h are not very helpful especially since it may depend on the specific range of parameters that are considered. Maybe 2D heat maps could be used, where each axis corresponds to a parameter and the color represents the noise reduction coefficient? Each controller could be represented by a different plot. c. P14, line 29. "A useful result is that the controller placement topology within the protein biosynthesis pathway can drastically alter the noise reduction ability of a controller. Particularly, we find that inhibition mediated via mRNA (MIX and MIL subtypes) is more efficacious than inhibition mediated by protein (PIX and PIL subtypes)". This is possibly an important result, and I find it surprising. Can the authors explain why this is the case?

1.7 P.12, lines15-16. "...using OR system is barely beneficial to local-MIL and is deleterious to NCR-MIL". Why? Can the authors propose a mechanistic explanation for this?

1.8 Presentation. a. Many figures are missing labels. For example, what is the unit of time in 1c and 1e? What is the unit of the y axis in 1c and 1e and of both axes in Fig. 1d & f? b. Many figure assignments are wrong or confusing. Examples include: * P5, line 30 (Figs. 1a & b do not show stochastic simulations or distributions as the sentence suggests) * p5, line 38 (Fig. 1g, not 1c) * p8, line 36 as well as others in the same paragraph c. Figures are most commonly referenced by using parentheses instead of "as shown in Fig. X and Y" as used by the authors. I suggest authors write, for example, "The GFP distribution is narrower with resource competition (Fig. XX)". d. The authors are using the present tense rather than the past tense to describe experiments/simulations done in the past.

Reviewer #2

The manuscript reports the role of resource competition (RNAP and ribosomes) in gene expression noise and how it can be controlled. The authors show that resource competition results in less noise arising from mRNA production, but introduces a new noise term coined as "resource competitive noise". They show that the overall gene expression noise with resource sharing is lower than a model with unlimited resources. They show that through the use orthogonal resources, noise reduction is retained while removing the added resource competitive noise. They study how different feedback control strategies reduce noise and conclude that global feedback controller can increase noise while the use of a local controller + orthogonal resources reduces noise. This work contributes towards better understanding noise prorogation due to resource sharing and how it can be mitigated. The problem formulation is clearly stated and sufficient background and motivation is provided. Major and minor comments follow.

Major Comments:

2.1 Is there a physical reason why the total noise is lower for the RC models vs the UR? Does this result depend on the specific parameters used for the simulation? Can it be proven mathematically?

2.2 J_p seems like a key parameter in the study. However, a precise mathematical definition was hard to find. Perhaps this can be explicitly defined in the main text along with Equation 1.

2.3 Is the dependence of RC in J_p through $H_2,3$? if so, it may help to explicitly show this dependence.

2.4 Similarly, how does m depend on J_p for the RC model? Through $H_2,1$? Why does it make sense that m monotonically increases with J_p , but is constant for the UR system? Not clear what "continuous relaxation of the resource limitations on translation" means.

2.5 The authors claim that resource competition noise is eliminated by using orthogonal resources. However, orthogonal and endogenous ribosomes are composed of the same r-proteins. If this was taken into account in the modeling, I'd expect correlation in the gene expression noise in figure 2-b. Should this be taken into account?

2.6 For the analysis of the feedback systems would it be possible to prove noise reduction analytically in a similar way as done for the orthogonal resource competition? Right now it is not clear if the results only hold for the simulated parameters or are general.

2.7 For each plot it would help to know what specific equations were simulated and with what parameters.

Minor comments:

2m1 Call out of supplementary figures (eg., page 7 line 6) is missing an indication that you are referring to supplementary material and makes reading the paper extremely difficult.

2m2 Fig. 1: Did not introduce RC in the caption. Even reading the figure description in the main text makes it difficult to comprehend. Is not until one arrives to equation 1, that is becomes more clear what RC is. Perhaps it's best introduce the equation earlier.

2m3 Introduction: Better motivation why we want noise control strategies. Is this currently an issue?

2m4 Are there experimental data in the references that support the authors findings? For example, in the RFP vs GFP experiments was anti-correlation observed?

2m5 Introduction: Are there any applications where gene expression noise be exploited? If so, perhaps this could be briefly mentioned.

2m6 Do the results from the paper hold for any organism? My guess is that the results may be specific to bacteria? If so, should this be mentioned anywhere.

Reviewer #3

The authors in this paper publish a new approach for modeling, analyzing, and attenuating noise in synthetic gene circuits. Specifically, they consider the problem of attenuating noise arising from scarce resources in the cell. This is an important topic, as noise invariably is present in single cell dynamics, but more so with genes that achieve a low copy number of expression. I believe the paper is aimed in the correct direction, but it requires significant work before it is in a publishable state. For this reason, I recommend a major revision. I would be happy to review the revision. I think it can be published eventually.

Concerns:

3.1: In single cells, molecular dynamics are best modeled as stochastic due to the discrete nature of molecular reactions and the low copy number of many reactions. These properties then raise an important question regarding the copy number of critical resources in the cell, that I believe should be more thoroughly addressed. Are resources in the cell highly susceptible to stochastic fluctuations? Typically enzymatic machinery and ribosomal machinery are present at 2-3 orders of magnitude higher than low copy genes. With such a high copy number, is noise a significant concern in host machinery? Further, most synthetic gene circuits are implemented on the genome (see the work of Voigt, Jewett, Murray, Arkin, Collins, etc.) to ensure genetic stability of the construct and minimal metabolic load. Given the stratification between the copy number of genes on the genome and copy numbers of ribosomal genes, it's not clear whether the noise levels are comparable between the two and whether the noise of ribosomal or transcriptional machinery is significant enough to propagate noise in the ways the authors describe (see Figure 1 Panels c-d). I believe this issue must be thoroughly investigated, as it is a fundamental premise as to **why this work is important**. If there is little noise propagating through resource constraints, then a fluctuation dissipation approach may not be suitable.

3.2: The presentation of the Figures leaves much to be desired. For example, Figure 1 C&E show single trajectories of circuit response in the presence and absence of resource competition. What are we to take away from these plots? Single cell trajectories from Gillespie simulations are almost always **random**, so the value of these plots is unclear. In plots Fig. 1 D,F, the takeaway is again unclear. It appears there is a correlation or skewing of what would be a symmetric distribution in the presence of η_{RC} (never defined in the Figure itself or the caption), but there is no salient increase in noise. The one plot that struck me as clear, but surprising, given the goals of the paper was Figure 1G. It appears the noise profile **barely changes** in simulation studies

between GFP expression with and without noisy resource competition. What is the point of the paper if there is hardly a difference? Since this is Figure 1, my overall impression was that the paper was not properly motivated, from a technical standpoint. Showing flow cytometry data or single cell RNAseq data that validated the premises of noise propagation through resource constraints seems like a critical first step. I have similar concerns about Figure 2D & 2B. Their purpose in the narrative of the paper is not clear. With Figure 2C, it is clear that applying an orthogonal resource results in a correction of the skewed bivariate distribution. Why are we not focusing on the decoupling of the GFP and RFP distributions rather than noise control? Perhaps this is an obvious conclusion, but again, where is the salient effect of noise control? Again Figure 2D seems to hardly register a difference between both schemes. Such a change in a flow cytometry readout would be nearly impossible to detect, if these were the two distributions. It would certainly be unrigorous to conclude that noise had been "controlled".

3.3: My next concern is that the noise does not appear to be controlled in any way that is significant. Specifically, the strongest conclusions on noise control arise from calculations from fluctuation dissipation analysts (which overall appears to be sound and rigorous on its own), but the actual simulations of systems subject to the conditions described in the schematics reveal little or minimal noise reduction. This is a major disconnect between theory and simulation that must be rectified before the paper can be published.

3.4: Are we assuming that dCas9 is not one of the resources being competed over? Further, wouldn't expression of dCas9 from synthetic promoters likewise produce a similar resource competition effect or load from the transcriptional and translational machinery? In this regard, I think the arguments of Paulsson and Vinnicombe's seminal paper need to be taken into account. As an independent peer (I am not either of those authors) the paper is not even cited; Paulsson and Vinnicombe's paper should be a foundational basis for the analysis in this work and a focal point of rebuttal. That paper should be a starting point for this paper - as it derives fundamental limits of noise suppression in molecular fluctuations! I look forward to reading the revision and seeing how the two works (this one and the Paulsson paper) are contrasted. Here is the link: <https://www.nature.com/articles/nature09333>

3.5: Figure 5 shows results that appear to be evaluations of expressions obtained from FDT analysis. This is fine. But a rigorous Gillespie simulation validating the degree of noise control asserted needs to be presented, including a table of the simulation parameters documenting what numerical assumptions were made. More generally, it is difficult to ascertain how the authors obtained their simulation results. In order for this paper to be publishable, the results of the simulation should be in a state of documentation that the work could be reproduced.

Overall, I think the paper centers on a very interesting topic. The authors clearly have a functional framework for analysis (FDT) and have made significant contributions in this regard. But there is a disconnect between theory and simulation. Further, it's not clear to me whether this situation would be experimentally relevant. Utilizing whole cell data that quantifies ribosomal and transcriptional copy numbers, their noise, and showing how it acts as a conduit for interaction between circuit components seems like a critical missing component to this paper. I highly recommend this paper for a major revision, followed by re-review. I strongly believe that this paper can ultimately be published in *Advanced Genetics*, after a major revision

Reviewer comments	Editor recommendation
3.4 Paulsson and Vinnicombe's paper .. should be a starting point for this paper - as it derives fundamental limits of noise suppression in molecular fluctuations!	ED0 please discuss this work in relation to your assumptions before introducing your results.
2.1 Is there a physical reason why the total noise is lower for the RC models vs the UR? Does this result depend on the specific parameters used for the simulation? 3.3 actual simulations of systems subject to the conditions described in the schematics reveal little or minimal noise reduction. This is a major disconnect between theory and simulation that must be rectified 3.5 a rigorous Gillespie simulation validating the degree of noise control asserted needs to be presented, including a table of the simulation parameters documenting what numerical assumptions were made.	ED1 provide evidence that this observation holds for a range of parameter values, or explain mathematically why this occurs. Are the assumptions of FDA justified, or do other simulations predict different amounts of noise propagation?
1.1 The distribution with RC may even be slightly wider, but the widths (full width at half maximum) were not quantified and compared. 1.4 There is generally a lack of quantification in the manuscript.	ED2 provide relevant measurements to support comparisons made - and statistics, for instance coefficients of variation
2m2 Fig. 1: Did not introduce RC in the caption. 2.2 J_p seems like a key parameter in the study. However, a precise mathematical definition was hard to find. 2.7 For each plot it would help to know what specific equations were simulated and with what parameters.	ED3 provide definitions and explanations for the parameters, assumptions and concepts at first use and alongside the equations.
3.1 Given the stratification between the copy number of genes on the genome and copy numbers of ribosomal genes, it's not clear whether the noise levels are comparable between the two and whether the noise of ribosomal or transcriptional machinery is significant enough to propagate noise in the ways the authors describe before the paper can be published.	ED4 use literature and simulations to explain how noise might be able to propagate despite very different copy numbers of genes and ribosomes.

Reviewer #1

In this manuscript, Goetz et al. present modeling results focusing on understanding how resource competition (RC) impacts the noise of genetic circuits. They also explore how different noise-reduction controllers perform in the presence of resource competition and when the synthetic proteins are expressed using orthogonal expression machinery. This is an interesting study, and while results haven't been confirmed experimentally, it could help guide the design of future synthetic circuits. The manuscript is generally easy to read and the figures are easy to understand. However, before being ready for publication, the authors should address the concerns listed below.

Response: We sincerely thank the Reviewer for appreciating the importance of this theoretical work for guiding the design of future synthetic gene circuits. We also give thanks for the invaluable comments, which have helped us to improve the manuscript significantly. Below, we have addressed all the comments and concerns raised by the Reviewer. All the changes in the manuscript and Supplementary Information are highlighted in blue.

Major concerns:

1.1 A key point of the paper is that RC both reduces and increases noise and the balance of these two properties results in a nonmonotonic dependence of the total noise level on resource availability. Indeed, this point is central to the paper as it defines its title. The authors mention that "We... find that resource competition narrows the GFP distribution, as shown in Fig. 1c". I assume that the authors meant 1g (distribution) and not 1c (stochastic trajectories). If indeed the graph the authors reference is 1g, the GFP

distributions are nearly the same between RC and non-RC conditions. The distribution with RC may even be slightly wider, but the widths (full width at half maximum) were not quantified and compared. Regardless of which distribution is wider, the difference is so small that I am not yet convinced of its importance.

Response: Thank the Reviewer for this critical comment. The figure reference in our latex file did not work properly somehow after we changed the format before submission, leading to the incorrect reference of supplementary figures as main figures. We apologize for the confusion and have corrected all these in the revised manuscript. For this claim, we wanted to refer to Supplementary Fig. 1c, in which we compare the RC and UR cases. But we do realize that the distribution difference between the systems with and without RC noise in Fig. 1g is not significant, so we did more analyses (see below).

The most pressing issue that the reviewer notes is the lack of significant difference in the distributions between the RC, UR, and OR cases demonstrated in Fig. 1g, Fig. 2d, and Sup Fig.2c. From these distributions, the effects of either RC or OR seem trivial. We had included these figures to demonstrate the existence of RC noise and had not optimized the parameters to demonstrate the extent to which these phenomena may occur. Particularly, our original parameter set resulted in only a small fraction of the overall noise to be due to resource competitive noise (RC noise). Following the Reviewer's comment, we did additional analyses and found that the resource competitive noise fraction depends primarily on the parameters translational rate k_p and translational capacity J_p . As shown in the new figure (Fig. S1a), RC noise fraction increases with the translational rate up to more than 30%. Interestingly, it shows a nonmonotonic dependence on J_p with a maximum at the middle (new Fig. S1a). In addition, we also did more stochastic simulation and analyzed the full width at half maximum following the Reviewer's suggestion. We

also found a similar peak at moderate J_p (new Fig. S1b) that is consistent with Fig.S1a. The underlying reason for the low RC noise fraction is that resource competition is not significant with a large value of J_p (high resource level), or the noise from the birth/death of protein (first noise term) is more dominant due to the low copy number of proteins with a small value of J_p (high resource level). Utilizing this information, we adjusted our test parameters to better demonstrate the extent of RC noise. In the revised manuscript, we changed these two parameters for all the simulations and updated all the figures. We found that now the RC noise is more significant, the GFP distribution shows a clear difference in Fig. 1g, 2d, and S2c. We believe these differences in the

Fig. S1a. Percentage of noise belonging to resource competition at the different protein production rates (k_p) as translational capacity (J_p) increases. (S1b) Width at half-height of probability distributions for the RC case as translational capacity (J_p) increases.

distributions are significant enough to properly demonstrate the extent of RC noise and the impact of applying orthogonal ribosomal systems.

The controllers, such as the MIX and MIL subtypes, introduced in the second part of the manuscript, perform significantly better at noise reduction (seen best in Fig. 3c,e,&g). In the revised manuscript, we have added an additional supplementary figure that has become the new Fig. S1 and updated all corresponding figures. We added the following sentences on Page 6, "*It is worthy to note that RC noise fraction actually depends on the translational rate k_p and the resource capacity J_p . As shown in Supplementary Figure 1a, the RC noise fraction increases with the translational rate up to more than 30%. Interestingly, it shows a nonmonotonic dependence on J_p with a maximum at the middle, consistent with the dependence of the full width at half maximum of the GFP distribution on J_p (Supplementary Figure 1b). The underlying reason for the low RC noise fraction at high or low J_p is that resource competition is not significant with a large value of J_p (high resource level), while the noise from the birth/death of protein (first noise term) is more dominant due to the low copy number of proteins with a small value of J_p (high resource level).*"

1.2 Figure 1f suggests that GFP and RFP expression isn't correlated in the non-RC case. However, Fig. 1e shows clear anticorrelations. The relative shape of GFP and RFP expression is so striking that the anticorrelations are unlikely be a visual artifact. Can the authors explain the discrepancy?

Response: Thanks for pointing out this discrepancy, which is due to a mistake in finalizing the figures before submission. We updated this figure, which now shows limited anticorrelation with a correlation coefficient of -0.01, suggesting no correlation.

1.3 Where are the values for the parameters used in the model e.g., mRNA degradation rate, transcription and translation rates, etc.? GFP and RFP have different folding/maturation properties; is that in the model? I don't think that the differences in properties between GFP and RFP are important for this story. Given this, why not just model "Protein 1 (P1)" and "Protein 2 (P2)"? If you want to keep using GFP and RFP so simplify the communication of the work, please clarify that they have identical properties in the model.

Response: Thank the Reviewer for this great suggestion. We did not consider the folding/maturation difference between the proteins. They are just simply used for the communication of the work as mentioned by the Reviewer. At the beginning of the Supporting Information Section A, we state that the two genes are identically regulated and that GFP/RFP is used simply to show result differences between the two genes. To make this clearer, we have added one more sentence here "*To differentiate between the two genes, fluorescent protein is used for the sole purpose of distinguishing protein expression.*" Additionally, this has been added in the description of the Results section on Page 3, "*Here, for simplicity, we considered the identical genes with same transcription/translation rates and mRNA/protein degradation rates, and did not consider the folding/maturation difference between the proteins and simply used them for the communication of the work.*"

1.4 There is generally a lack of quantification in the manuscript. For example, on p.8 (first paragraph), the authors say "Accordingly, the 2D GFP/RFP probability distribution is narrower than the RC case", but no quantification (coefficient of variation) is provided. Another example is in the previous sentence, where correlations/anticorrelations are not quantified. Please rectify this for the entire manuscript.

Response: We thank the Reviewer for this suggestion and have added quantification to the manuscript. To account for the absence of noise quantification, a new plot (Fig. S1b) was added, allowing for quantification of probability distribution to be seen much clearer. In addition, Fig. 1h and Fig. S2d (previously Fig. S1d) now

Fig. 1e. Gillespie stochastic trajectories of GFP (green) and RFP (red) expression with correlation coefficient = -0.01.

have a vertical line representing the J_p value used in Fig.1c-g and Fig. S2a-c to indicate the coefficient of variation and the three components of the total noise. Additionally, the correlation coefficient for Fig. 1c is -0.79 and has been added in the main text and figure caption. The correlation coefficient for Fig. 1e is -0.01 and has been added in the main text and figure caption. The correlation coefficient for new Fig. S2 (previously Fig. S1a) is -0.05 and has been added in the main text and figure caption.

1.5 Fig. 2e. Why does GFP noise increase with RFP expression in the OR case? Shouldn't their expression (and therefore their noise) be orthogonal? Is it an artifact of the scaling method used? If so, I'm not convinced that these results are useful.

Response: Thanks for pointing this out. Yes, this is due to the parameter scaling. We wanted to use the same scaling method for all the figures using RC case as the base parameter. We now agree with the Reviewer that this could cause some unnecessary confusion by using this scaling method in this figure and have changed to use the OR case as the base for Fig. 2 and Fig. S3 (originally Fig.S2) in the revised manuscript. We updated Figs. 2e, which now shows even more significant noise reduction by the OR system. We updated Fig.S3a-b (previously Fig. S2b-c), which now shows GFP total noise and the two terms making up this noise do not change with RFP mean in the OR system, but in the RC system GFP total noise increase with RFP mean because both the noises from the fluctuation of two mRNAs (η_m and η_{RC}) increase although the noise from the birth/death of GFP (η_p) is a constant. We also removed Fig. S2a and revised the descriptions of Fig. 2 and S3 accordingly. We have revised the description of these figures, pages 7-8, 'We then studied how the GFP noise levels in the RC and OR systems change with the RFP mean by increasing the effective RFP gene copy number. By rescaling the parameters in the RC system with OR system as the base, we make the means of the mRNAs and proteins in the RC system the same as the OR system. We found that, when there is no RFP, the noise levels in the two systems are equal, as shown by the red dot in Figure 2e. As the RFP mean increases, the GFP noise level in the OR system is constant given that its two components does not change with RFP mean (Supplementary Figure 3a), but in RC system the total noise increases because both the noises from the fluctuation of two mRNAs (η_m and η_{RC}) increase although the noise from the birth/death of GFP (η_p) is a constant (Supplementary Figure 3b), thus making the difference in noise level between two systems increase with RFP mean (Figure 2e). Note that the maximum RFP mean reached by increasing the copy number is due to the saturation mediated by resource limitation, as shown in Supplementary Figure 3c. This conclusion holds true if the plasmid copy numbers increase for both genes. The total noise decreases and reaches a saturation floor, as shown in Figure 2f, which is consistent with the previous findings [34-37]. Nonetheless, the noise levels in the UR system are always the largest, suggesting that utilizing orthogonal resources is a good strategy to eliminate the contribution of resource competition to gene expression noise.'

(2e) FDT analysis revealing the dependence of the GFP total noise levels on RFP mean.

(S3a) The dependence of the GFP total noise level and its two decompositions over RFP mean in the OR system. (S3b) The dependence of the GFP total noise level and its three decompositions over RFP mean in the RC system.

1.6 It is not clear whether the differences between controllers (MIL, PIL, MIX, and PIX) are due to differences in their design or the value of specific parameters. a. How robust are the results to their parameters? b. Average plots like Fig. 4g/h are not very helpful especially since it may depend on the specific range of parameters that are considered. Maybe 2D heat maps could be used, where each axis corresponds to a parameter and the color represents the noise reduction coefficient? Each controller could be represented by a different plot. c. P14, line 29. "A useful result is that the controller placement topology within the protein biosynthesis pathway can drastically alter the noise reduction ability of a controller. Particularly, we find that inhibition mediated via mRNA (MIX and MIL subtypes) is more efficacious than inhibition mediated by protein (PIX and PIL subtypes)". This is possibly an important result, and I find it surprising. Can the authors explain why this is the case?

Response: We agree with the Reviewer that the 2D heatmaps are good to be used to show the noise reductions across two parameters. We have many heatmaps to show the dependence of the noise reductions on two key parameters, the controller strength S_c and the load of the competing module RFP mean. We want to emphasize that the range of S_c and RFP means are not set arbitrarily. They both start with 0 to represent no controller or no competing module. The maximum of S_c was set to the value when the noise reduction reaches saturation, and the maximum of R2 was set to the value when the RFP mean reaches its saturation. The average plot in Fig. 4g-h was used to find out which control placement typology works better by averaging all the three controller types across the GC, LC, and NCR controllers.

The reason why MIX and MIL are more efficacious than PIX and PIL is that proteins have multiple noise resources, including its birth/death process, the fluctuation from the translation step, and the noise propagated from the opposing mRNA, as shown in our analytical solution, while the mRNA noise only depends on its birth/death. In this way, protein has more noise resources than mRNA, which would cause more variance if they were the effectors for the controllers. In the revised manuscript, we have added this discussion for this claim on page 14, '*The reason why MIX and MIL are more efficacious than PIX and PIL is that proteins have three resources of noise while the mRNA noise only depends on its birth/death. In this way, protein has more noise resources than mRNA, which would cause more variance if they were the effectors for the controllers.*'

1.7 P.12, lines15-16. "...using OR system is barely beneficial to local-MIL and is deleterious to NCR-MIL". Why? Can the authors propose a mechanistic explanation for this?

Response: Mechanistically, in NCR MIL, when the control node from GFP mRNA inhibits its translation, it also promotes the translation of RFP no matter if RFP level is above or under its average in the OR case, thus could increase RFP noise. In the RC case, on the other hand, it only promotes the translation of RFP when RFP is under its average, thus decreasing RFP noise. That is, NCR is designed for combating the resources competition, so it may not work well in the OR case. For the MIL, NCR already works as the best, so these effects lead to the results that the OR system is barely beneficial to local-MIL and is deleterious to NCR-MIL. For the MIX, because NCR does not work very well, so the OR system can further enhance both the NCR-MIX and LC-MIX. In the revised manuscript, we have added this discussion on page 11-12, '*The underlying reason is that for the NCR-MIL controller, when the control node from RFP mRNA inhibits its translation, it also promotes the translation of GFP no matter if GFP level is above or under its average in the OR case, thus could actually increase GFP noise. In the RC case, on the other hand, it only promotes the translation of GFP when GFP is under its average thus decreasing GFP noise. That is, NCR is designed for combating the resources competition, so it may not work well with OR system. The GFP noise normalized to the OR case without a controller reveals that the addition of a local-MIX or NCR-MIX controller is consistently beneficial for the OR system in further attenuating noise, as shown in Figure 5c and Supplementary Figures. 7d-7f.*'

1.8 Presentation. a. Many figures are missing labels. For example, what is the unit of time in 1c and 1e? What is the unit of the y axis in 1c and 1e and of both axes in Fig. 1d & f? b. Many figure assignments are wrong or confusing. Examples include: * P5, line 30 (Figs. 1a & b do not show stochastic simulations or distributions as the sentence suggests) * p5, line 38 (Fig. 1g, not 1c) * p8, line 36 as well as others in the same paragraph c.

Figures are most commonly referenced by using parentheses instead of "as shown in Fig. X and Y" as used by the authors. I suggest authors write, for example, "The GFP distribution is narrower with resource competition (Fig. XX)". d. The authors are using the present tense rather than the past tense to describe experiments/simulations done in the past.

Response: We had added "hours" as the unit of time for all stochastic simulations and replaced the y-axis label of "Protein Levels" with "Protein Numbers" for clarification. The figure reference in our latex file did not work properly somehow after we changed the format before submission, leading to the incorrect reference of supplementary figures as main figures. We apologize for the confusion, and we have corrected all these in the revised manuscript. In the main text, "Fig. 1a & b" has been relabeled as Supplementary Fig. 2a & b (previously Fig. S1a&b), "Fig 1c" has been relabeled as Supplementary Fig 2c (previously Fig. S1c), and "Figures 1h and 1d" has been relabeled as Fig. 1h and Supplementary Fig. 2d (previously Fig. S1d). Grammatical errors involving past/present tense for experiments and simulations have been fixed so that the reader can tell the difference between what the lab did during the experiment versus the results of the experiment.

Reviewer #2

The manuscript reports the role of resource competition (RNAP and ribosomes) in gene expression noise and how it can be controlled. The authors show that resource competition results in less noise arising from mRNA production, but introduces a new noise term coined as "resource competitive noise". They show that the overall gene expression noise with resource sharing is lower than a model with unlimited resources. They show that through the use orthogonal resources, noise reduction is retained while removing the added resource competitive noise. They study how different feedback control strategies reduce noise and conclude that global feedback controller can increase noise while the use of a local controller + orthogonal resources reduces noise. This work contributes towards better understanding noise prorogation due to resource sharing and how it can be mitigated. The problem formulation is clearly stated and sufficient background and motivation is provided. Major and minor comments follow.

Response: We sincerely thank the Reviewer for appreciating the importance of the study for understanding noise prorogation due to resource competition and its mitigation. Following the Reviewer's valuable suggestions and comments, we have significantly improved the manuscript. Please find the point-by-point response below. All the changes in the manuscript and Supplementary Information are highlighted in blue.

Major Comments:

2.1 Is there a physical reason why the total noise is lower for the RC models vs the UR? Does this result depend on the specific parameters used for the simulation? Can it be proven mathematically?

Response: We thank the Reviewer for this insightful comment. In Fig. S2d (Original Fig. S1d), we showed that total noise is lower for the RC case vs the UR. Mathematically, it is also true. Here is the proof:

Total noise in the RC case,

$$\eta_{p_1}^2 \text{ total} = \frac{\sigma_{p_1}^2}{P_1^2} = \frac{1}{P_1} + \frac{\sigma_{m_1}^2}{M_1^2} \times H_{21}^2 \times \frac{1}{\left(\frac{1}{\tau_1} + \frac{1}{\tau_2}\right)} + \frac{\sigma_{m_2}^2}{M_2^2} \times H_{23}^2 \times \frac{1}{\left(\frac{1}{\tau_3} + \frac{1}{\tau_2}\right)},$$

Total noise in the UR case,

$$\eta_{p_1}^2 \text{ total} = \frac{\sigma_{p_1}^2}{P_1^2} = \frac{1}{P_1} + \frac{\sigma_{m_1}^2}{M_1^2} \times H_{21}^2 \times \frac{1}{\left(\frac{1}{\tau_1} + \frac{1}{\tau_2}\right)}$$

We scaled the parameters so that we have the same mean numbers of proteins and mRNA in both cases. The first noise term is the same as it only depends on the protein number. The mRNA noise in the 2nd and/or 3rd term, which depends on the mean number of two mRNAs, are also the same. In addition, the time scale parameters in the 2nd and/or 3rd terms are the same for two identical genes. Thus we only need to compare the difference between the values of $H_{21}^2 + H_{23}^2$ in RC case and H_{21}^2 in the UR case. In the UR case,

$$H_{21}^2 = 1$$

In the RC case, we have

$$\begin{aligned} H_{21}^2 + H_{23}^2 &= \left(\frac{m_2 + J_p \Omega}{m_1 + m_2 + J_p \Omega} \right)^2 + \left(\frac{m_2}{m_1 + m_2 + J_p \Omega} \right)^2 = \left(\frac{m_2 + J_p \Omega}{m_1 + m_2 + J_p \Omega} \right)^2 + \left(\frac{m_1}{m_1 + m_2 + J_p \Omega} \right)^2 \\ &< \left(\frac{m_1 + m_2 + J_p \Omega}{m_1 + m_2 + J_p \Omega} \right)^2 = 1 \end{aligned}$$

Thus, the total noise in the RC case is always smaller than UR case. In the revised manuscript, we also have added this proof in the Supporting Information Section E and added some description in the main text, Page 6, "*It can be seen that GFP noise is always smaller in the RC case, which is also proved mathematically (Supporting Information Section E).*"

2.2 J_p seems like a key parameter in the study. However, a precise mathematical definition was hard to find. Perhaps this can be explicitly defined in the main text along with Equation 1.

Response: Thanks for this great suggestion. The mathematical definition of J_m and J_p is from our previous work [5]. We have then generally described them in the Supporting Information Section A as "the effective transcriptional and translational capacities of limited resources in the host cell for synthetic gene circuits, respectively." To add clarification for the readers, we have added the mathematical definition of J_p by referring to our previous work [6] in the Supporting Information Section A.

2.3 Is the dependence of RC in J_p through $H_{2,3}$? if so, it may help to explicitly show this dependence.

Response: We agree with the Reviewer that it helps to explicitly show how the η_{RC} depends on J_p through H_{23} ,

$$\eta_{RC} = \frac{\sigma_{m_2}^2}{M_2^2} \times H_{23}^2 \times \frac{\frac{1}{\tau_2}}{\left(\frac{1}{\tau_3} + \frac{1}{\tau_2} \right)}$$

where $H_{23} = \frac{\partial \ln(J_{p_1}^-/J_{p_1}^+)}{\partial \ln(M_2)}$, with $J_{p_1}^-$ and $J_{p_1}^+$ as

$$\begin{aligned} J_{p_1}^- &= d_{p_1} P \\ J_{p_1}^+ &= \frac{k_{p_1} M_1 \Omega}{1 + \sum \frac{M_j / \Omega}{J_{p_j}}} \end{aligned}$$

We analytical get

$$H_{23} = \frac{J_{p_1} m_2}{J_{p_1} m_2 + J_{p_2} m_1 + J_{p_1} J_{p_2} \Omega}$$

If $J_{p_1} = J_{p_2} = J_p$

$$H_{23} = \frac{m_2}{m_1 + m_2 + J_p \Omega}$$

Thus η_{RC} decreases with J_p , as shown in Fig. 1h.

2.4 Similarly, how does η_m depend on J_p for the RC model? Through H_{21} ? Why does it make sense that η_m monotonically increases with J_p , but is constant for the UR system? Not clear what "continuous relaxation of the resource limitations on translation" means.

Response: Same here, the η_m depends on J_p through H_{21} ,

$$\eta_m = \frac{\sigma_{m_1}^2}{M_1^2} \times H_{21}^2 \times \frac{\frac{1}{\tau_2}}{\left(\frac{1}{\tau_1} + \frac{1}{\tau_2}\right)}$$

where $H_{21} = \frac{\partial \ln(J_{p_1}^-/J_{p_1}^+)}{\partial \ln(M_1)}$ with $J_{p_1}^-$ and $J_{p_1}^+$ defined as

$$J_{p_1}^- = d_{p_1}P$$

$$J_{p_1}^+ = \frac{k_{p_1}M_1\Omega}{1 + \sum \frac{M_j/\Omega}{J_{p_j}}}$$

We analytical get

$$H_{21} = -\frac{J_{p_1}(m_2 + J_{p_2}\Omega)}{J_{p_1}m_2 + J_{p_2}m_1 + J_{p_1}J_{p_2}\Omega}$$

If $J_{p_1} = J_{p_2} = J_p$

$$H_{21} = -\frac{m_2 + J_p\Omega}{m_1 + m_2 + J_p\Omega}$$

Thus η_m monotonically increases with J_p , as shown in Fig. 1h.

For the UR system,

$$J_{p_1}^- = d_{p_1}P$$

$$J_{p_1}^+ = k_{p_1}M_1\Omega.$$

We analytical get

$$H_{21} = -1$$

Thus, η_m does not change with J_p , as shown in Sup Fig. 2d.

In the revised manuscript, we have added the analytical expression of H_{23} and H_{21} in both RC and UR systems in the Supporting Information Section D (Page 6-8) and referred to them in the main text for explaining how η_{RC} and η_m depend on J_p in the RC and UR system. Page 6, *'It is also noted that η_{RC} and η_m depend on J_p through H_{23} and H_{21} , which decrease and increase with J_p respectively (Eq. 33 and Eq.34), as shown in Figure 1h'.*

2.5 The authors claim that resource competition noise is eliminated by using orthogonal resources. However, orthogonal and endogenous ribosomes are composed of the same r-proteins. If this was taken into account in the modeling, I'd expect correlation in the gene expression noise in figure 2-b. Should this be taken into account?

Response: Thank the Reviewer for this critical comment. We completely understand that we still do not have a completely orthogonal system in the field now. The orthogonal system published in the field still used some components in the host system. For example, orthogonal and endogenous ribosomes are composed of the same r-proteins as mentioned by the Reviewer. We did not consider the potential resource competition by the orthogonal control system but simply assumed that it needs much fewer resources compared with the synthetic gene circuits. In other words, our work gives the maximum bound of the noise reduction by the orthogonal system. In the revised manuscript, we have added the following sentence in the discussion part:

Page 7, *'Here, we assume that the orthogonal system uses much less resource than the synthetic gene circuit and thus did not consider the potential resource competition by the orthogonal system.'*

2.6 For the analysis of the feedback systems would it be possible to prove noise reduction analytically in a similar way as done for the orthogonal resource competition? Right now it is not clear if the results only hold for the simulated parameters or are general.

Response: It is impossible to solve for the noise expression for the feedback systems analytically. We understand that the numerical simulation depends on the parameters, so we did not set the range of Sc and RFP means arbitrarily. They both start with 0 to represent no controller or no competing module. The maximum of Sc was set to the value when the noise reduction reaches saturation, and the maximum of $R2$ was set to the value when the RFP mean reaches its saturation. In the revised manuscript, we have added these sentences on page 9, 'To compare these controllers fairly, both Sc and RFP start with 0 to represent no controller or no competing module. The maximum of Sc was set to the value when the noise reduction reaches saturation, and the maximum value of $R2$ was set to the value when the RFP mean reaches its saturation.'

In addition, we have added mechanical explanations for key findings following Reviewer 1's comments. For example, to explain why using OR system is barely beneficial to local-MIL and is deleterious to NCR-MIL, we have added the following sentences on pages 11-12, 'The underlying reason is that for the NCR-MIL controller, when the control node from GFP mRNA inhibits its translation, it also promotes the translation of RFP no matter if RFP level is above or under its average in the OR case, thus could actually increase RFP noise. In the RC case, on the other hand, it only promotes the translation of RFP when RFP is under its average thus decreasing RFP noise. That is, NCR is designed for combating the resources competition, so it may not work well with OR system.'

2.7 For each plot it would help to know what specific equations were simulated and with what parameters.

Response: In the Supporting Information Section I, "Base parameters and parameter rescale," all parameters are stated along with the range used for parameters when parameters are varied within a simulation. To make sure readers can clearly identify which equations were used for which simulations, we added which figures used each specific equation set in the Supporting Information Sections.

Minor comments:

2m1 Call out of supplementary figures (eg., page 7 line 6) is missing an indication that you are referring to supplementary material and makes reading the paper extremely difficult.

Response: The figure reference in our latex file did not work somehow after we changed the format before submission, leading to the incorrect reference of supplementary figures as main figures. We apologize for the confusion, and we have corrected all these in the revised manuscript. In the revised manuscript, "Fig. 1a & b" has been relabeled as Supplementary Fig. 2a & b (previously Fig. S1a&b), "Fig 1c" has been relabeled as Supplementary Fig 2c (previously Fig. S1c), "Figures 1h and 1d" has been relabeled as Fig. 1h and Supplementary Fig. 2d (previously Fig. S1d).

2m2 Fig. 1: Did not introduce RC in the caption. Even reading the figure description in the main text makes it difficult to comprehend. Is not until one arrives to equation 1, that is becomes more clear what RC is. Perhaps it's best introduce the equation earlier.

Response: The first use of the abbreviation "RC" appears on Page XX Line XX immediately following the phrase "resource competition." To add clarification, we have added a description to the captions of Fig. 1a. We guess the Reviewer may talk about RC noise (η_{RC}). We did not introduce the equation earlier as we tried to show the existence with stochastic simulation and probability distribution first. However, in the revised manuscript, we moved the definition of RC noise (η_{RC}) to the beginning of that paragraph before we described Fig. 1e-f, as well as in the figure caption and Method section.

Page 4, 'We defined the noise from the fluctuation of the opposing mRNA due to resource competition as resource competitive noise (RC noise), denoted as η_{RC} '.

2m3 Introduction: Better motivation why we want noise control strategies. Is this currently an issue?

Response: In the revised manuscript, we have added this discussion in the Introduction section on page 2, 'Noise is one of the fundamental factors that limit the performance of synthetic gene circuits. Such impairment of function has been noted as far back as the repressilator circuit [7], where fewer than half of the cells showed highly variable oscillations with large variability in period and amplitude. Gene expression noise is one of the sources of uncertainty that can lead to circuit failure [8]. That is, noise, as one of the notorious issues, reduces the forward engineerability of synthetic gene circuits and impairs circuit performance.'

2m4 Are there experimental data in the references that support the authors findings? For example, in the RFP vs GFP experiments was anticorrelation observed?

Response: Resource-competition mediated anticorrelation and mutual inhibition have been noted in the field previously, and the impact of resource-competitive phenomena has gained much attention recently. Several publications have observed an anticorrelation of two gene/modules [3, 5, 13, 18]. Gyorgy et al. demonstrated that a trade-off between genetic components could be described via isocost lines, with RBS strength determining the magnitude of the negative slope between the two genes' expression levels and plasmid copy number determining the vertical shift of the isocost line. In a previous publication from our lab, we demonstrated that resource competitive anticorrelation could lead to winner-takes-all behavior [5]. In the revised manuscript, we have added this discussion when we describe Fig. 1c-d on page4, 'This is consistent with the recent finding of the anticorrelation between two independently regulated gene/modules [4, 19, 14, 6].'

2m5 Introduction: Are there any applications where gene expression noise be exploited? If so, perhaps this could be briefly mentioned.

Response: We agree with the Reviewer that the gene expression noise could be exploited under some context. In the revised manuscript, we have briefly mentioned this in the Discussion:

Page 15, 'It is noted that gene expression noise could be exploited even it is undesirable for the deterministic functions of gene circuits. For example, gene expression noise induces a bimodal response in a positive feedback loop circuit without cooperativity [72]. At the populational level, stochastic phenotype switching due to the gene expression noise gives bacteria an advantage in fluctuating extracellular environments [73, 74, 75, 76, 77].'

2m6 Do the results from the paper hold for any organism? My guess is that the results may be specific to bacteria? If so, should this be mentioned anywhere.

Response: The results from this paper hold true generally for any system wherein two identical genetic modules stochastically compete over ribosomal or transcriptional resources to an extent, as our model is a very general model on the central dogma of protein expression and significant resource competition is observed in both bacteria and mammalian cells. However, the exact resource competitive parameters most likely vary by organism. Of particular interest and importance is the distribution of resource competitive phenomena between the transcriptional and translational steps; it remains a topic of interest in the study of resource competition of whether competition over RNA polymerases or competition over ribosomal moieties play a stronger role in defining resource competitive behavior for several notable model organisms. Furthermore, combined competition between general and other more system-specific factors such as sigma factors and system-specific ribosomal units may also vary by the organism in ways that begin to deviate from our model to an extent. In the

revised manuscript, we have added this discussion on page 14, 'The conclusions from this work theoretically apply to any organism as long as there exists significant resource competition, which is already observed in both bacteria and mammalian cells [4, 19, 14, 6]'

Reviewer #3

The authors in this paper publish a new approach for modeling, analyzing, and attenuating noise in synthetic gene circuits. Specifically, they consider the problem of attenuating noise arising from scarce resources in the cell. This is an important topic, as noise invariably is present in single cell dynamics, but more so with genes that achieve a low copy number of expression. I believe the paper is aimed in the correct direction, but it requires significant work before it is in a publishable state. For this reason, I recommend a major revision. I would be happy to review the revision. I think it can be published eventually.

Response: We sincerely thank the Reviewer for appreciating the importance of our manuscript and for the valuable comments. Below, we have addressed all the comments and questions raised by the Reviewer. All the changes in the manuscript and Supplementary Information are highlighted in blue.

Concerns:

3.1: In single cells, molecular dynamics are best modeled as stochastic due to the discrete nature of molecular reactions and the low copy number of many reactions. These properties then raise an important question regarding the copy number of critical resources in the cell, that I believe should be more thoroughly addressed. Are resources in the cell highly susceptible to stochastic fluctuations? Typically enzymatic machinery and ribosomal machinery are present at 2-3 orders of magnitude higher than low copy genes. With such a high copy number, is noise a significant concern in host machinery? Further, most synthetic gene circuits are implemented on the genome (see the work of Voigt, Jewett, Murray, Arkin, Collins, etc.) to ensure genetic stability of the construct and minimal metabolic load. Given the stratification between the copy number of genes on the genome and copy numbers of ribosomal genes, it's not clear whether the noise levels are comparable between the two and whether the noise of ribosomal or transcriptional machinery is significant enough to propagate noise in the ways the authors describe (see Figure 1 Panels c-d). I believe this issue must be thoroughly investigated, as it is a fundamental premise as to *why this work is important*. If there is little noise propagating through resource constraints, then a fluctuation dissipation approach may not be suitable.

Response: We thank the Reviewer for this critical comment. We agree with the Reviewer that enzymatic machinery and ribosomal machinery are present at 2-3 orders of magnitude higher than low copy genes. However, expression of the synthetic gene circuits indeed causes a significant burden to the host cells, and resource competition indeed causes problems, such as the significantly reduced expression of one gene due to the induction of another gene [4, 6, 14, 19] and Winner-Takes-All problem between two self-activation modules [6]. Here we actually did not consider the fluctuation of the resource levels but instead focused on how the fluctuation expression of one gene affects the other gene expression with a fixed number of limited resources. In the last paragraph, we have this discussion on Page 15, 'In our work, potential sources of extrinsic noise in the system such as the fluctuation of the copy numbers of the transcriptional resource (RNAPs) and translation resource (ribosomes) are not included.'

We agree with the Reviewer that if the synthetic gene circuits are implemented as a single copy on the genome, the resource competition is probably not a significant problem. While not all synthetic gene circuits work with a single copy on the genome, many circuits need a high copy number of plasmids. For example, in our previous work [6], we tried to solve the winner-takes-all resource competition problem by using low copy number plasmid and found that cell states are already not well separated, and the resource competition is still a problem. Thus, while we are still using plasmid as one important approach for implementing synthetic gene circuits, we need to consider the resource competition into our circuit design and understand how the noise behavior is changed due to resource competition and develop controlling strategies to mitigate these effects. It is worthy to note that resource competition is actually now well recognized in the field, as seen in many recent publications (ref. [1-6]).

In the revised manuscript, we have added this discussion on page 3, 'While the competition is not significant with only one copy of the two genes [22], it can become significant for a system with high-copy plasmid that are sometime are very necessary for the circuit functions.'

3.2: The presentation of the Figures **leaves much to be desired**. For example, Figure 1 C&E show single trajectories of circuit response in the presence and absence of resource competition. What are we to take away from these plots? Single cell trajectories from Gillespie simulations are almost always *random*, so the value of these plots is unclear. In plots Fig. 1 D,F, the takeaway is again unclear. It appears there is a correlation or skewing of what would be a symmetric distribution in the presence of η_{RC} (never defined in the Figure itself or the caption), but there is no salient increase in noise. The one plot that struck me as clear, but surprising, given the goals of the paper was Figure 1G. It appears the noise profile *barely changes* in simulation studies between GFP expression with and without noisy resource competition. What is the point of the paper if there is hardly a difference? Since this is Figure 1, my overall impression was that the paper was not properly motivated, from a technical standpoint. Showing flow cytometry data or single cell RNAseq data that validated the premises of noise propagation through resource constraints seems like a critical first step. I have similar concerns about Figure 2D & 2B. Their purpose in the narrative of the paper is not clear. With Figure 2C, it is clear that applying an orthogonal resource results in a correction of the skewed bivariate distribution. Why are we not focusing on the decoupling of the GFP and RFP distributions rather than noise control? Perhaps this is an obvious conclusion, but again, where is the salient effect of noise control? Again Figure 2D seems to hardly register a difference between both schemes. Such a change in a flow cytometry readout would be nearly impossible to detect, if these were the two distributions. It would certainly be unrigorous to conclude that noise had been "controlled".

Response: We thank the Reviewer for these insightful comments. The most pressing issue that the reviewer notes is the lack of significant difference in the distributions in between the RC, UR, and OR cases demonstrated in Fig. 1g, Fig. 2d, and Sup Fig.2c. From these distributions, the effects of either RC or OR seem trivial. We had included these figures to demonstrate the existence of RC noise and had not optimized our parameters to demonstrate the extent to which these phenomena may occur. Particularly, our original parameter set resulted in only a small fraction of resource competitive noise (RC noise). Following the Reviewer's comment, we did additional analyses and found that the resource competitive noise fraction depends primarily on the parameters translational rate k_p and translational capacity J_p . As shown in the new figure (Fig. S1a), RC noise fraction increases with the translational rate up to more than 30%. Interestingly, it shows a nonmonotonic dependence on J_p with a maximum at the middle (new Fig. S1a). The underlying reason for the low RC noise fraction is that resource competition is not significant with a large value of J_p (high resource level), while the noise from the birth/death of protein (first noise term) is more dominant due to the low copy number of proteins with a small value of J_p (high resource level). Utilizing this information, we adjusted our test parameters to better demonstrate the extent of RC noise. In the revised manuscript, we changed these two parameters for all the simulations and updated all the figures. We found that now the RC noise is more significant, the GFP distribution shows a clear difference in Fig. 1g, 2d, and S2c. We believe these differences in the distributions are significant enough to properly demonstrate the extent of RC noise and the impact of applying orthogonal ribosomal systems.

In the revised manuscript, we moved the definition of RC noise (η_{RC}) to the beginning of that paragraph before we described Fig. 1e-f, as well as in the figure caption and Method section. Page 4, 'We defined the noise from the fluctuation of the opposing mRNA due to resource competition as resource competitive noise (RC noise), denoted as η_{RC} '.

With regards to the figures demonstrating stochastic Gillespie simulation trajectories, Figure 1C and 1E, we had included these trajectories to demonstrate specifically the anti-correlatory nature of resource competitive noise as compared to the noncorrelated nature of the more conventional sources of genetic noise noted previously. Of note, due to a mistake in finalizing Figure 1E, both the trajectory with and without resource competitive noise to

appear anticorrelated. This issue has been rectified. Similarly, the intended purpose of including Figure 2b was to demonstrate that the application of orthogonal ribosomal systems is capable of nullifying the anticorrelatory noise resulting from the resource competitive term.

We agree with the Reviewer that decoupling of GFP and RFP is a very important strategy. In the field, several strategies are proposed to decouple them, including the OR system and negative feedback loop. However, no prior works have studied noise behavior. This is one of the motivations of this study. It is noted that these control strategies have some limitations. For example, the OR system itself also takes additional resources in the host cell, so it is challenging to build up an idea OR system, so we need to also consider other control strategies such as feedback controllers. In our previous work [20], we showed that the local and global controllers proposed in the field are not as good as our proposed NCR controller for the winner-take-all resource competition. Here we want to compare whether they work similarly for noise control. In the revised manuscript, we have added this discussion on page 13, '*Current OR systems are not completely orthogonal to the host system and take up additional resources in the host cell, thus is necessary to also consider other control strategies such as feedback controllers to further improve the control of resource competition.*'

3.3: My next concern is that the noise does not appear to be controlled in any way that is significant. Specifically, the strongest conclusions on noise control arise from calculations from fluctuation dissipation analysts (which overall appears to be sound and rigorous on its own), but the actual simulations of systems subject to the conditions described in the schematics reveal little or minimal noise reduction. This is a major disconnect between theory and simulation that must be rectified before the paper can be published.

Response: Our analysis was originally conducted to demonstrate that the negative feedback controllers presented could offer noise reduction capabilities. However, originally we had not optimized our parameter set to demonstrate the extent to which this was possible. Particularly, our original parameter set actually led to only a low fraction of the RC noise for the controllers to reduce. Exploring the dependence of resource competitive noise fraction on parameters, we found that the RC noise fraction depends strongly on translational capacity J_p and translational rate constant k_p and optimized our parameters accordingly to give a test system with a high fraction of RC noise. As a larger fraction of the noise is now RC noise, applying our controllers to this new test system resulted in significantly increased noise reduction efficacies, with our best controllers demonstrating more than 50% noise reduction as seen in the updated Fig 3c-d, 4d, 5a-b. This large noise reduction is verified not only by our FDT analysis but also via rigorous Gillespie simulations that we have added to the manuscript to confirm the results of our FDT in Supplementary Fig. 9.

3.4: Are we assuming that dCas9 is not one of the resources being competed over? Further, wouldn't expression of dCas9 from synthetic promoters likewise produce a similar resource competition effect or load from the transcriptional and translational machinery? In this regard, I think the arguments of Paulsson and Vinnicombe's seminal paper need to be taken into account. As an independent peer (I am not either of those authors) the paper is not even cited; **Paulsson and Vinnicombe's paper should be a foundational basis for the analysis in this work and a focal point of rebuttal.** That paper should be a starting point for this paper - as it derives fundamental limits of noise suppression in molecular fluctuations! I look forward to reading the revision and seeing how the two works (this one and the Paulsson paper) are contrasted. Here is the link: <https://www.nature.com/articles/nature09333>

Response to Reviewer: In the NCR design, dCas9 is one of the resources that the controller loops of two modules compete over. This is the most important difference between the local and NCR controllers. In the local controller, the two controller loops of two modules are independent, while in NCR, they are connected by competing for the limited dCas9. In this way, the NCR controller not only punishes the module that takes relatively too much resource but also gives an advantage to the other module at the same time. In this work, we did not consider the resource competition by expressing dCas9 as the Reviewer mentioned. The reason is that dCas9 is designed to

be limited for the negative competition in the NCR controller design. That is, we do not want dCas9 to be expressed at a high level. Otherwise, it will be like the local controllers. For future experimental design, a tunable dCas9 system will be integrated into the genome instead of the synthetic gene circuit plasmid. In the revised manuscript, we have added the following sentences in the

Page 9, 'Here we did not consider the resource competition by expressing dCas9 as we need the dCas9 to be limited so that the two modules could compete over to achieve the negative competition as designed in the NCR controller. For future experimental design, a tunable dCas9 system will be integrated into the genome instead of the synthetic gene circuit plasmid.'

We completely understand the Reviewer's comment that Paulsson and Vinnicombe's 2010 Nature paper set a fundamental limitation to noise suppression through negative feedback loops. While this limitation is the same for all the controllers, here we focused on studying if the noise reduction depends on the controller topology under the context of resource competition, which is a foundational problem in the field of synthetic biology. In future work, we will revise our mathematical modeling framework by considering the resource competition by the controller components and further study whether these controllers are able to reach the limitation of noise suppression. In the revised manuscript, we have cited Paulsson and Vinnicombe's 2010 Nature and 2019 PRL papers ([49]) and added the following sentences in the Introduction,

Page 8, 'Some theoretical analyses have provided us a fundamental limitation to the noise suppression through negative feedback loops [48, 49]. While cellular systems in nature may have evolved complicated regulatory networks to operate close to this fundamental limit, it is still unclear whether different types of negative feedback controllers perform similarly for noise reduction of the synthetic gene circuits, especially under the context of resource competition.'

3.5: Figure 5 shows results that appear to be evaluations of expressions obtained from FDT analysis. This is fine. But a rigorous Gillespie simulation validating the degree of noise control asserted needs to be presented, including a table of the simulation parameters documenting what numerical assumptions were made. More generally, it is difficult to ascertain how the authors obtained their simulation results. In order for this paper to be publishable, the results of the simulation should be in a state of documentation that the work could be reproduced.

Response: To support the findings garnered from our FDT analysis, we have added a new figure (Fig. S9) detailing rigorous Gillespie simulation for validating the conclusions of our FDT analysis. The results of these simulations coincide with all the important conclusions of our FDT analysis. They support our finding that RC is capable of reducing noise with respect to the UR case, and OR is capable of reducing the noise even further. They support the majority of the controller topologies utilized are able to significantly reduce noise, with controllers found to be most efficacious from the FDT analysis also outperforming others in the Gillespie simulations. In the revised manuscript, we have added this discussion on page 11, *'This is also validated by the Gillespie simulation (Figure 9)'*.

Overall, I think the paper centers on a very interesting topic. The authors clearly have a functional framework for analysis (FDT) and have made significant contributions in this regard. But there is a disconnect between theory and simulation. Further, it's not clear to me whether this situation would be experimentally relevant. Utilizing whole cell data that quantifies ribosomal and transcriptional copy numbers, their noise, and showing how it acts

as a conduit for interaction between circuit components seems like a critical missing component to this paper. I highly recommend this paper for a major revision, followed by re-review. I strongly believe that this paper can ultimately be published in Advanced Genetics, after a major revision.

Response to Reviewer: Thank the Reviewer again for appreciating the importance of the work and our contribution to this interesting topic. Also thank the Reviewer for believing that this work could be published in Advanced Genetics after major revision. We agree with the Reviewer that it would be more convincing all the conclusion prediction by the theoretical analyses by quantifying the ribosomal and transcriptional copy numbers, their noise experimentally. While we are not able to do this in our lab and there is no published data to directly prove that the circuit components indirectly interact due to the limited shared resources, we want to point it out that resource competition is already indirectly proved in many synthetic gene circuits systems including our [1-6], where we could easily see that the gene circuit modules indirectly inhibit each other due to resource competition.

2nd Peer Review	12/20/2021 to 1/4/2022
-------------------------------

Reviewer #2

The authors have addressed all my questions and concerns. I would read the manuscript closely one last time for consistency (there were a few minor typos and errors in figure/equation referencing).

2nd Editorial Decision	1/5/2022
--	-----------------

I am pleased to inform you that your manuscript has been accepted in principle for publication pending a clean final version correcting remaining typographic errors and references. I invite you to respond to the reviewer comments and make the necessary changes to your manuscript.

Final Decision	1/10/2022
------------------

The manuscript is formally accepted for publication.